# Estimating and Controlling for Equalized Odds via Sensitive Attribute Predictors

**Beepul Bharti**
Johns Hopkins University
bbharti1@jhu.edu

**Paul Yi**
University of Maryland
pyi@som.umaryland.edu

**Jeremias Sulam**
Johns Hopkins University
jsulam1@jhu.edu

## Abstract

As the use of machine learning models in real world high-stakes decision settings continues to grow, it is highly important that we are able to audit and control for any potential fairness violations these models may exhibit towards certain groups. To do so, one naturally requires access to sensitive attributes, such as demographics, biological sex, or other potentially sensitive features that determine group membership. Unfortunately, in many settings, this information is often unavailable. In this work we study the well known *equalized odds* (EOD) definition of fairness. In a setting without sensitive attributes, we first provide tight and computable upper bounds for the EOD violation of a predictor. These bounds precisely reflect the worst possible EOD violation. Second, we demonstrate how one can provably control the worst-case EOD by a new post-processing correction method. Our results characterize when directly controlling for EOD with respect to the predicted sensitive attributes is – and when is not – optimal when it comes to controlling worst-case EOD. Our results hold under assumptions that are milder than previous works, and we illustrate these results with experiments on synthetic and real datasets.

## 1 Introduction

Machine learning (ML) algorithms are increasingly used in high stakes prediction applications that can significantly impact society. For example, ML models have been used to detect breast cancer in mammograms [37], inform parole and sentencing decisions [14], and aid in loan approval decisions [41]. While these algorithms often demonstrate excellent overall performance, they can be dangerously unfair and negatively impact under-represented groups [38]. Some of these unforeseen negative consequences can even be fatal, e.g. deep learning models for chest x-ray disease classification exhibit under diagnosis bias towards certain sensitive groups [36]. Recommendations to ensure that ML systems do not exacerbate societal biases have been raised by several groups, including the White House, which in 2016, released a report on big data, algorithms, and civil rights [31]. It is thus critical to understand how to rigorously evaluate the fairness of ML algorithms and control for unfairness during model development.

These needs have prompted considerable research in the area of *Fair ML*. While there exist many definitions of algorithmic fairness [7], common notions of *group fairness* consider different error rates of a predictor across different groups: males and females, white and non-white, etc. For example, the *equal opportunity* criterion requires the true positive rate (TPR) be equal across both groups, while *equalized odds* requires both TPR and false positive rate (FPR) to be the same across groups [21]. Obtaining predictors that are fair therefore requires enforcing these constraints on error rates across groups during model development, which can be posed as a constrained (or regularized) optimization problem [35, 42, 2, 13, 16]. Alternatively, one can devise post-processing strategies to modify a certain predictor to correct for differences in TPR and FPR [21, 18, 12, 1], or even include data

37th Conference on Neural Information Processing Systems (NeurIPS 2023).

pre-processing steps that ensure that unfair models could not be obtained from such data to begin with [39, 8].

Naturally, all these techniques for estimating or enforcing fairness require access to a dataset with features, $X$, responses, $Y$, and sensitive attributes, $A$. However, in many settings this is difficult or impossible, as datasets often do not include samples that have all these variables. This could be because the sensitive attribute data was withheld due to privacy concerns, which is very common with medical data due to HIPAA federal law requirements, or simply because it was deemed unnecessary to record [40, 43]. A real-world example of this is in the recent Kaggle-hosted RSNA Chest X-Ray Pneumonia Detection Challenge [33]. Even though this dataset of chest x-rays was painstakingly annotated for pneumonia disease by dozens of radiologists, it did not include sensitive attributes (e.g., age, sex, and race), precluding the evaluation of fairness of the models developed as part of the challenge. In settings like this, where there is limited or no information on the sensitive attribute of interest, it is still important to be able to accurately estimate the violation of fairness constraints by a ML classifier and to be able to alleviate these violations before deploying it in a sensitive application

This leads to the natural question:

*How can one assess and control the fairness of a classifier without having access to sensitive attribute data?*

In other words, how can we measure and potentially control the fairness violations of a classifier for $Y$ with respect to a sensitive attribute $A$, when we have no data that jointly observes $A$ and $Y$?

## 1.1 Related Work

Estimating unfairness and, more importantly, developing fair predictors where there is no – or only partial – information about the sensitive attribute has only recently received increasing attention. The recent work by Zhao et al. [44] explores the perspective of employing features that are correlated with the sensitive attribute, and shows that enforcing low correlation with such "fairness related features" can lead to models with lower bias. Although these techniques are promising, they require domain expertise to determine which features are highly correlated with the sensitive attribute. An appealing alternative that has been studied is the use proxy sensitive attributes that are created by a second predictor trained on a different data set that contains only sensitive attribute information [20, 25, 10, 5]. This strategy has been widely adopted in many domains such as healthcare [17], finance [6], and politics [23]. While using sensitive attribute predictors has proven to be an effective and practical solution, it must be done with care, as this opens new problems for estimating and controlling for fairness. The work by Prost et al. [34] considers the estimation of fairness in a setting where one develops a predictor for an unobserved covariate, but it does not contemplate predicting the sensitive attribute itself. On the other hand Chen et al. [10] study the sources of error in the estimation of fairness via predicted proxies computed using threshold functions, which are prone to over-estimation.

The closest to our work are the recent results by Kallus et al. [25], and Awasthi et al. [5, 4]. Kallus et al. [25] study the identifiability of fairness violations under general assumptions on the distribution and classifiers. They show that, in the absence of the sensitive attribute, the fairness violation of predictions, $\widehat{Y}$, is unidentifiable unless strict assumptions [1] are made or if there is some common observed data over $A$ and $Y$. Nonetheless, they show not all hope is lost and provide closed form upper and lower bounds of the fairness violations of $\widehat{Y}$ under the assumption that one has two datasets: one that is drawn from the marginal over $(X, A)$ and the other drawn from the marginal over $(X, Y, \widehat{Y})$. Their analysis, however, does not consider predictors $\widehat{Y} = f(X)$ or $\widehat{A} = h(X)$, and instead their bounds depend explicitly on the conditional probabilities, $\mathbb{P}(A \mid X)$ and $\mathbb{P}(\widehat{Y}, Y \mid X)$, along with the distribution over the features, $\mathbb{P}(X)$. Unfortunately, with this formulation, it is unclear how the bounds would change if the estimation of the conditional probabilities was inaccurate (as is the case when developing predictors $\widehat{A}$ in the real world). Furthermore, in settings where $X$ is high dimensional (as for image data), calculating such bounds would become intractable. Since the fairness violation cannot be directly modified, they then study when these bounds can be reduced and improved. However, they do so in settings that impose smoothness assumptions over $(X, A, Y, \widehat{Y})$,

---

[1]Kallus et al. [25] show that if, $\widehat{Y}, Y \perp\!\!\!\perp A \mid X$, then the fairness violations are identifiable.

which clearly are not-verifiable without data over the complete joint distribution. As a result, their results do not provide any actionable method that could improve the bounds.

Awasthi et al. [5], on the other hand, make progress in understanding properties of the sensitive attribute predictor, $\widehat{A}$, that are desirable for easy and accurate fairness violation estimation and control of a classifier $\widehat{Y}$. Assuming that $\widehat{Y} \perp\!\!\!\perp \widehat{A} \mid (A, Y)$, they demonstrate that the true fairness violation is in fact proportional to the estimated fairness violation (the fairness violation using $\widehat{A}$ in lieu of A). This relationships yields the counter-intuitive result that given a fixed error budget for the sensitive attribute predictor, the optimal attribute predictor for the estimation of the true fairness violation is one with the most unequal distribution of errors across the subgroups of the sensitive attribute. However, one is still unable to actually calculate the true fairness violation – as it is unidentifiable. Nonetheless, the relationship does demonstrates that if one can maintain the assumption above while controlling for fairness with respect to $\widehat{A}$, then doing so will provably reduce the true fairness violation with respect to $A$. Unfortunately, while these rather strict assumptions can be met in some limited scenarios (as in [4]), these are not applicable in general – and cannot even be tested without access to data over $(A, Y)$.

Overall, while progress has been made in understanding how to estimate and control fairness violations in the presence of incomplete sensitive attribute information, these previous results highlight that this can only be done in simple settings (e.g., having access to some data from the entire distribution, or by making strong assumptions of conditional independence). Moreover, it remains unclear whether tight bounds can be obtained that explicitly depend on the properties of the predictor $\widehat{Y}$, allowing for actionable bounds that can provably mitigate for its fairness violation without having an observable sensitive attribute or making stringent assumptions.

## 1.2 Contributions

The contributions of our work can be summarized as follows:

- We study the well known equalized odds (EOD) definition of fairness in a setting where the sensitive attributes, $A$, are not observed with the features $X$ and labels $Y$. We provide tight and computable bounds on the EOD violation of a classifier, $\widehat{Y} = f(X)$. These bounds represent the *worst-case* EOD violation of $f$ and employ a predictor for the sensitive attributes, $\widehat{A} = h(X)$, obtained from a sample over the distribution $(X, A)$.

- We provide a precise characterization of the classifiers that achieve *minimal worst-case* EOD violations with respect to unobserved sensitive attributes. Through this characterization, we demonstrate *when* simply correcting for fairness with respect to the proxy sensitive attributes will yield *minimal worst-case* EOD violations, and *when* instead it proves to be sub-optimal.

- We provide a simple and practical post-processing technique that provably yields classifiers that maximize prediction power while achieving *minimal worst-case EOD violations* with respect to unobserved sensitive attributes.

- We illustrate our results on a series of simulated and real data of increasing complexity.

## 2 Problem Setting

We work within a binary classification setting and consider a distribution $\mathcal{Q}$ over $(\mathcal{X} \times \mathcal{A} \times \mathcal{Y})$ where $\mathcal{X} \subseteq \mathbb{R}^n$ is the feature space, $\mathcal{Y} = \{0, 1\}$ the label space, and $\mathcal{A} = \{0, 1\}$ the sensitive attribute space. Furthermore, and adopting the setting of [5], we consider 2 datasets, $\mathcal{D}_1$ and $\mathcal{D}_2$. The former is drawn from the marginal over $(\mathcal{X}, \mathcal{A})$ of $\mathcal{Q}$ while $\mathcal{D}_2$ is drawn from the marginal $(\mathcal{X}, \mathcal{Y})$ of $\mathcal{Q}$. In this way, $\mathcal{D}_1$ and $\mathcal{D}_2$ contain the same set of features, $\mathcal{D}_1$ contains sensitive attribute information and $\mathcal{D}_2$ contains label information. The drawn samples in $\mathcal{D}_1$ and $\mathcal{D}_2$ are i.i.d over their respective marginals and different from one another.

Similar to previous work [10, 34, 5], we place ourselves in a *demographically scarce* regime where there is designer who has access to $\mathcal{D}_1$ to train a sensitive attribute predictor $h : \mathcal{X} \rightarrow \mathcal{A}$ and a developer, who has access to $\mathcal{D}_2$, the sensitive attribute classifier $h$, and all computable probabilites $\mathbb{P}(h(X), A)$ that the designer of $h$ can extract from $\mathcal{D}_1$. In this setting, the goal of the developer is to learn a classifier $f : \mathcal{X} \rightarrow \mathcal{Y}$ (from $\mathcal{D}_2$) that is fair with respect to $A$ utilizing $\widehat{A}$. The central idea is

to augment every sample in $\mathcal{D}_2$, $(x_i, y_i)$, by $(x_i, y_i, \hat{a}_i)$, where $\hat{a}_i = h(x_i)$. Intuitively, if the error of the sensitive attribute predictor, denoted herein by $U = \mathbb{P}(h(X) \neq A)$, is low, we could hope that fairness with respect to the real (albeit unobserved) sensitive attribute can be faithfully estimated. Our goal is to thus estimate the error incurred in measuring and enforcing fairness constraints by means of $\hat{A} = h(X)$, and potentially alleviate or control for it.

Throughout the remainder of this work we focus on *equalized odds* (EOD) as our fairness metric [21] of interest as it is one of the most popular notions of fairness. Thus moving forward, the term fairness refers specifically to EOD. We denote $\hat{Y} = f(X)$ for simplicity, and for $i, j \in \{0, 1\}$ define the group conditional probabilities

$$\alpha_{i,j} = \mathbb{P}(\hat{Y} = 1 \mid A = i, Y = j). \tag{1}$$

These probabilities quantify the TPR (when $j = 1$) and FPR (when $j = 0$), for either protected group ($i = 0$ or $i = 1$). We assume that the base rates, $r_{i,j} = \mathbb{P}(A = i, Y = j) > 0$ so that these quantities are not undefined. With these conditionals probabilities, we define the true fairness violation of $f$, with the two quantities

$$\Delta_{\text{TPR}}(f) = \alpha_{1,1} - \alpha_{0,1} \quad \text{and} \quad \Delta_{\text{FPR}}(f) = \alpha_{1,0} - \alpha_{0,0} \tag{2}$$

which respectively quantify the absolute difference in TPR and FPRs among the two protected groups.

We also need to characterize the performance of the sensitive attribute classifier, $h$. The miss-classification error of $h$ can be decomposed as, $U = U_0 + U_1$ where $U_i = \mathbb{P}(\hat{A} = i, A \neq i)$, for $i \in \{0, 1\}$. We define the difference in errors to be

$$\Delta U = U_0 - U_1. \tag{3}$$

In a demographically scarce regime, the rates $r_{i,j}$, and more importantly the quantities of interest, $\Delta_{\text{TPR}}(f)$ and $\Delta_{\text{FPR}}(f)$, cannot be computed because samples from $A$ and $Y$ are not jointly observed. However, using the sensitive attribute classifier $h$, we can predict $\hat{A}$ on $\mathcal{D}_2$ and compute

$$\hat{r}_{i,j} = \mathbb{P}(\hat{A} = i, Y = j) \quad \text{and} \quad \hat{\alpha}_{i,j} = \mathbb{P}(\hat{Y} = 1 \mid \hat{A} = i, Y = j),$$

which serve as the estimates for the true base rates and group TPRs and FPRs.

## 3 Theoretical Results

With the setting defined, we will now present our results. The first result provides computable bounds on the true fairness violation of $f$ with respect to the true, but unobserved, sensitive attribute $A$. The bounds precisely characterize the *worst-case* fairness violation of $f$. Importantly, as we will explain later, this first result will provide insight into what properties $f$ must satisfy so that its worst-case fairness violation is *minimal*. In turn, these results will lead to a simple post-processing method that can correct a pretrained classifier $f$ into another one, $\bar{f}$, that has minimal worst-case fairness violations. Before presenting our findings, we first describe the key underlying assumption we make about the pair of classifiers, $h$ and $f$, so that the subsequent results are true.

**Assumption 1.** *For $i, j \in \{0, 1\}$, the classifiers $\hat{Y} = f(X)$ and $\hat{A} = h(X)$ satisfy*

$$\frac{U_i}{\hat{r}_{i,j}} \leq \hat{\alpha}_{i,j} \leq 1 - \frac{U_i}{\hat{r}_{i,j}}. \tag{4}$$

To parse this assumption, it is easy to show that this is met when a) $h$ is accurate enough for the setting, namely that $\mathbb{P}(\hat{A} = i, A \neq i) \leq \frac{1}{2}\mathbb{P}(\hat{A} = i, Y = j)$, and b) the predictive power of $h$ is better than the ability of $f$ to predict the labels, $Y$ – or more precisely, $\mathbb{P}(\hat{A} = i, A \neq i) \leq \mathbb{P}(\hat{Y} = j, \hat{A} = i, Y \neq j)$. We refer the reader to Appendix A for a thorough explanation for why this is true. While this assumption may seem limiting, this is milder than those in existing results: First, accurate predictors $h$ can be developed [6, 17, 23, 19], thus satisfying the assumption (our numerical results will highlight this fact as well). Second, other works [5, 9], require assumptions on $\hat{A}$ and $\hat{Y}$ that are unverifiable in a demographically scarce regime. Our assumption, on the other hand, can always be easily verified because all the quantities are computable. Furthermore, this assumption can be also be relaxed if one desires partial guarantees. More specifically, if the above is true for $i \in \{0, 1\}$ and only for $j = 1$, then all subsequent results for $\Delta_{\text{TPR}}$ hold. Similarly, if it only holds for $j = 0$, the results for $\Delta_{\text{FPR}}$ hold. Therefore, if one only cares about the equal opportunity definition of fairness, as an example, then this only needs to hold for $i \in \{0, 1\}$ and $j = 1$.

## 3.1 Bounding Fairness Violations with Proxy Sensitive Attributes

With Assumption 1 in place, we present our main result.

**Theorem 1** (Bounds on $\Delta_{\text{TPR}}(f)$ and $\Delta_{\text{FPR}}(f)$). *Under Assumption 1, we have that*

$$
|\Delta_{TPR}(f)| \leq B_{TPR}(f) \triangleq \max\{|B_1 + C_{0,1}|, |B_1 - C_{1,1}|\}
$$
$$
|\Delta_{FPR}(f)| \leq B_{FPR}(f) \triangleq \max\{|B_0 + C_{0,0}|, |B_0 - C_{1,0}|\}
$$

(5)

*where*

$$
B_j = \frac{\hat{r}_{1,j}}{\hat{r}_{1,j} + \Delta U}\widehat{\alpha}_{1,j} - \frac{\hat{r}_{0,j}}{\hat{r}_{0,j} - \Delta U}\widehat{\alpha}_{0,j} \quad and \quad C_{i,j} = U_i\left(\frac{1}{\hat{r}_{1,j} + \Delta U} + \frac{1}{\hat{r}_{0,j} - \Delta U}\right).
$$

*Furthermore, the upper bounds for $|\Delta_{TPR}(f)|$ and $|\Delta_{FPR}(f)|$ are tight.*

The proof, along with all others in this work, are included in Appendix A. We now make a few remarks on this result. First, the bound is tight in that there exists settings (albeit unlikely) with particular marginal distributions such that the bounds hold with equality. Second, even though $|\Delta_{\text{TPR}}(f)|$ and $|\Delta_{\text{FPR}}(f)|$ cannot be calculated, a developer can still calculate the *worst-case* fairness violations, $B_{\text{TPR}}(f)$ and $B_{\text{FPR}}(f)$, because these depend on quantities that are all computable in practice. Thus, if $B_{\text{TPR}}(f)$ and $B_{\text{FPR}}(f)$ are low, then the developer can proceed having a guarantee on the maximal fairness violation of $f$, even while not observing $|\Delta_{\text{TPR}}(f)|$ and $|\Delta_{\text{FPR}}(f)|$. On the other hand, if these bounds are large, this implies a *potentially* large fairness violation over the protected group $A$ by $f$ and the developer should not proceed in deploying $f$ and instead seek to learn another classifier with smaller bounds. Third, the obtained bounds are linear in the parameters $\widehat{\alpha}_{i,j}$, which the developer *can adjust* as they are properties of $f$: this will become useful shortly.

## 3.2 Optimal Worst-Case Fairness Violations

Given the result above, what properties should classifiers $f$ satisfy such that $B_{\text{TPR}}(f)$ and $B_{\text{FPR}}(f)$ are minimal? Moreover, are the classifiers $f$ that are fair with respect to $\widehat{A}$, the ones that have *smallest* $B_{\text{TPR}}(f)$ and $B_{\text{FPR}}(f)$? We now answer these questions in the following theorem.

**Theorem 2** (Minimizers of $B_{\text{TPR}}(f)$ and $B_{\text{FPR}}(f)$). *Let $\widehat{A} = h(X)$ be a fixed sensitive attribute classifier with errors $U_0$ and $U_1$ that produces rates $\hat{r}_{i,j} = \mathbb{P}(\widehat{A} = i, Y = j)$. Let $\mathcal{F}$ be the set of all predictors of $Y$, parameterized by rates $\widehat{\alpha}_{i,j}$, that, paired with $h$, satisfy Assumption 1. Then, $\exists \overline{Y} \in \mathcal{F}$ with group conditional probabilities, $\widehat{\underline{\alpha}}_{i,j} = \mathbb{P}(\overline{Y} = 1 \mid \widehat{A} = i, Y = j)$ that satisfies the following condition,*

$$
\frac{\hat{r}_{0,j}}{\hat{r}_{0,j} - \Delta U}\widehat{\underline{\alpha}}_{0,j} - \frac{\hat{r}_{1,j}}{\hat{r}_{1,j} + \Delta U}\widehat{\underline{\alpha}}_{1,j} = \frac{\Delta U}{2}\left(\frac{1}{\hat{r}_{1,j} + \Delta U} + \frac{1}{\hat{r}_{0,j} - \Delta U}\right).
$$

(6)

*Furthermore, any such $\overline{Y}$ has minimal maximal fairness violation, i.e. $\forall \widehat{Y} \in \mathcal{F}$,*

$$
|\Delta_{TPR}(\overline{Y})| \leq B_{TPR}(\overline{Y}) \leq B_{TPR}(\widehat{Y}) \quad and \quad |\Delta_{FPR}(\overline{Y})| \leq B_{FPR}(\overline{Y}) \leq B_{FPR}(\widehat{Y}).
$$

(7)

This result provides a precise characterization of the conditions that lead to minimal worst-case fairness violations. Observe that if $\Delta U \neq 0$, the classifier with minimal $B_{\text{TPR}}(f)$ and $B_{\text{FPR}}(f)$ involves $\widehat{\alpha}_{i,j}$ such that $\widehat{\alpha}_{1,j} \neq \widehat{\alpha}_{0,j}$, i.e. it is *not* fair with respect to $\widehat{A}$. On the other hand, if the errors of $h$ are balanced ($\Delta U = 0$), then minimal bounds are achieved by being fair with respect to $\widehat{A}$.

## 3.3 Controlling Fairness Violations with Proxy Sensitive Attributes

Now that we understand what conditions $f$ must satisfy so that it's worst case fairness violations are minimal, what remains is a method to obtain such a classifier. We take inspiration from the post-processing method proposed by Hardt et al. [21], which derives a classifier $\overline{Y} = \bar{f}(X)$ from $\widehat{Y} = f(X)$ that satisfies equalized odds with respect to a sensitive attribute $A$ while minimizing an expected missclassification loss – *only applicable if one has access to $A$*, which is not true in our setting. Nonetheless, since our method will generalize this idea, we first briefly comment on this

approach. The method they propose works as follows: given a sample with initial prediction $\widehat{Y} = \hat{y}$ and sensitive attribute $A = a$, the derived predictor $\bar{f}$, with group conditional probabilities $\underline{\alpha}_{i,j} = \mathbb{P}(\overline{Y} = 1 \mid A = i, Y = j)$, predicts $\overline{Y} = 1$ with probability $p_{a,\hat{y}} = \mathbb{P}(\overline{Y} = 1 \mid A = a, \widehat{Y} = \hat{y})$. The four probabilities $p_{0,0}, p_{0,1}, p_{1,0}, p_{1,1}$ can be then calculated so that $\overline{Y}$ satisfies equalized odds and the expected loss between $\overline{Y}$ and labels $Y$, i.e. $\mathbb{E}[L(\overline{Y}, Y)]$, is minimized. The fairness constraint, along with the objective to minimize the expected loss, give rise to the linear program:

**Equalized Odds Post-Processing (Hardt et al. [21])**

$$\min_{p_{a,\hat{y}} \in [0,1]} \quad \mathbb{E}[L(\overline{Y}, Y)] \quad \text{subject to} \quad \underline{\alpha}_{0,j} = \underline{\alpha}_{1,j} \quad \text{for} \quad j \in \{0,1\}. \tag{8}$$

Returning to our setting where we *do not* have access to $A$ but only proxy variables $\widehat{A} = h(X)$, we seek classifiers $f$ that are fair with respect to the sensitive attribute $A$. Since these attributes are not available, (thus rendering the fairness violation to be unidentifiable [25]), a natural alternative is to minimize the worst-case fair violation with respect to $A$, which *can* be computed as shown in Theorem 1. Of course, such an approach will only minimize the worst case fairness and one cannot certify that the true fairness violation will decrease because – as explained above – it is unidentifiable. Nonetheless, since we know what properties *optimal* classifiers must satisfy (as per Theorem 2), we can now modify the above problem to construct a corrected classifier, $\bar{f}$, as follows. First, we must employ $\widehat{A}$ in place of $A$, which amounts to employing $\widehat{\alpha}_{i,j}$ in lieu of $\alpha_{i,j}$. To this end, denote the (corrected) group conditional probabilities of $\overline{Y}$ to be $\underline{\widehat{\alpha}}_{i,j}$. Second, the equalized odds constraint is replaced with the constraint in Theorem 2. Lastly, we also enforce the additional constraints detailed in Assumption 1 on the $\underline{\widehat{\alpha}}_{i,j}$. With these modifications in place, we present the following generalized linear program:

**Worst-case Fairness Violation Reduction**

$$\min_{p_{\hat{a},\hat{y}} \in [0,1]} \quad \mathbb{E}[L(\overline{Y}, Y)]$$

subject to

$$\frac{\hat{r}_{0,j}}{\hat{r}_{0,j} + \Delta U} \underline{\widehat{\alpha}}_{0,j} - \frac{\hat{r}_{1,j}}{\hat{r}_{1,j} - \Delta U} \underline{\widehat{\alpha}}_{1,j} = \frac{\Delta U}{2} \left( \frac{1}{\hat{r}_{1,j} + \Delta U} + \frac{1}{\hat{r}_{0,j} - \Delta U} \right) \tag{9}$$

$$\frac{U_i}{\hat{r}_{i,j}} \leq \underline{\widehat{\alpha}}_{i,j} \leq 1 - \frac{U_i}{\hat{r}_{i,j}} \quad \text{for} \quad i, j \in \{0,1\}.$$

The solution to this linear program will yield a classifier $\bar{f}$ that satisfies Assumption 1, has minimal $B_{\text{TPR}}(\bar{f})$ and $B_{\text{FPR}}(\bar{f})$, and has minimal expected loss. Note, that if $\Delta U = 0$, then the coefficients of $\underline{\widehat{\alpha}}_{0,j}$ and $\underline{\widehat{\alpha}}_{1,j}$ will equal one, and so the first set of constraints simply reduces to $\underline{\widehat{\alpha}}_{0,j} = \underline{\widehat{\alpha}}_{1,j}$ for $j \in \{0,1\}$ and the linear program above is precisely the post-processing method of [21] with $\widehat{A}$ in place of $A$ (while enforcing Assumption 1).

Let us briefly recap the findings of this section: We have shown that in a demographically scarce regime, one can provide an upper bound on the true fairness violation of a classifier (Theorem 1). Second, we have presented a precise characterization of the classifiers with minimal worst-case fairness violations. Lastly, we have provided a simple and practical post-processing method (a linear program) that utilizes a sensitive attribute predictor to construct classifiers with minimal worst-case fairness violations with respect to the true, unknown, sensitive attribute.

## 4 Experimental Results

We now illustrate our theoretical results on synthetic and real world datasets. The code and data necessary to reproduce these experiments are available at `https://github.com/Sulam-Group/EOD-with-Proxies`.

### 4.1 Synthetic Data

We begin with a synthetic example that allows us to showcase different aspects of our results. The data consists of 3 features, $X_1, X_2, X_3 \in \mathbb{R}$, sensitive attribute $A \in \{0,1\}$, and response $Y \in \{0,1\}$.

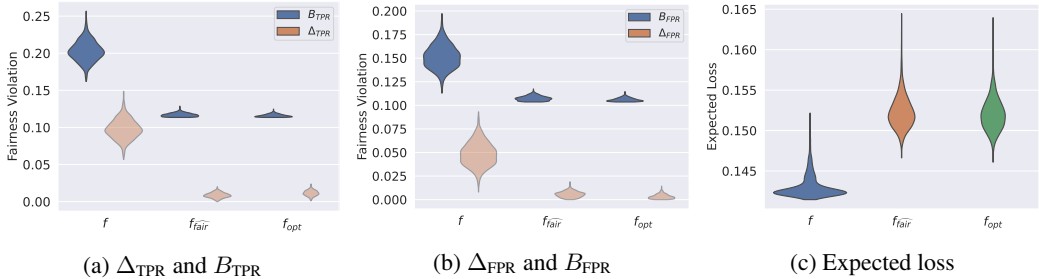

(a) $\Delta_{\text{TPR}}$ and $B_{\text{TPR}}$  (b) $\Delta_{\text{FPR}}$ and $B_{\text{FPR}}$  (c) Expected loss

Figure 1: Synthetic data ($\Delta U = 0$): true fairness violations, worst-case fairness violations, and expected loss for $f$, $f_{\widehat{\text{fair}}}$, and $f_{\text{opt}}$

The features are sampled from $(X_1, X_2, X_3) \sim \mathcal{N}(\mu, \Sigma)$, with

$$\mu = \begin{bmatrix} 1 \\ -1 \\ 0 \end{bmatrix} \quad \text{and} \quad \Sigma = \begin{bmatrix} 1 & 0.05 & 0 \\ 0.05 & 1 & 0 \\ 0 & 0 & 0.05 \end{bmatrix}.$$

The sensitive attribute, $A$, response $Y$, and classifier $f$, are modeled as

$$A = \mathbb{I}[(X_3 + 0.1) \geq 0]$$
$$Y \mid X, A \sim \text{Bernoulli}(S(X_1 + X_2 + X_3 + \epsilon_0(1 - A) + \epsilon_1 A))$$
$$f(X; c_1, c_2) = \mathbb{I}(S(c_1 X_1 + c_2 X_2 + c_3 X_3) \geq 0.5)$$

where $\mathbb{I}(\cdot)$ is the indicator function, $S(\cdot)$ is the sigmoid function and $c_1, c_2, c_3 \sim \mathcal{N}(1, 0.01)$. Furthermore, $\epsilon_0 \sim \mathcal{N}(0, 1), \epsilon_1 \sim \mathcal{N}(0, 0.5)$ are independent noise variables placed on the samples belonging to the groups $A = 0$ and $A = 1$ to ensure there is a non trivial fairness violation. Specifically, $\text{Var}(\epsilon_0) > \text{Var}(\epsilon_1)$ makes $f(X; c_1, c_2, c_3)$ unfair with respect to $A = 0$. Lastly, to measure the predictive capabilities of $f$, we use the loss function, $L(\hat{Y} \neq y, Y = y) = \mathbb{P}(Y \neq y)$, as this maximizes the well known Youden's Index[2].

### 4.1.1 Equal Errors ($\Delta U = 0$)

We model the sensitive attribute predictor as

$$h(X; \delta) = \mathbb{I}((X_3 + 0.1 + \delta) \geq 0)$$

where $\delta \sim \mathcal{N}(0, \sigma^2)$. We choose $\sigma^2$ so that $h(X)$ has a total error $U \approx 0.04$ distributed so that $\Delta U \approx 0$ with $U_0 \approx U_1 \approx 0.02$. We generate 1000 classifiers $f$ and for each one, calculate $\Delta_{\text{TPR}}, \Delta_{\text{FPR}}, B_{\text{TPR}}, B_{\text{FPR}}$, and $\mathbb{E}[L(f, Y)]$. Then, on each $f$, we run the (naïve) equalized odds post processing algorithm to correct for fairness with respect $\hat{A}$ to yield a classifier $f_{\widehat{\text{fair}}}$, and we also run our post-processing algorithm to yield an optimal classifier $f_{\text{opt}}$. For both sets of classifiers we again calculate the same quantities.

The results in Fig. 1 present the true fairness violations, worst-case fairness violations, and expected loss for the 3 sets of different classifiers. Observe that both $B_{\text{TPR}}$ and $B_{\text{FPR}}$ are significantly lower for $f_{\widehat{\text{fair}}}$ and $f_{\text{opt}}$ *and* that these values for both sets of classifiers are approximately the same. This is expected as $U_0 \approx U_1$ and so performing the fairness correction algorithm and our proposed algorithm amount to solving nearly identical linear programs. We also show the true fairness violations, $\Delta_{\text{TPR}}$ and $\Delta_{\text{FPR}}$ for all the classifiers to portray the gap between the bounds and the true values. As mentioned before, these true values *cannot be calculated* in a real demographically scarce regime. Nonetheless, the developer of $f$ now knows, post correction, that $|\Delta_{\text{TPR}}|, |\Delta_{\text{FPR}}| \lesssim 0.15$. Lastly, observe that the expected loss for $f_{\widehat{\text{fair}}}$ and $f_{\text{opt}}$ are naturally higher compared to that of $f$, however the increase in loss is minimal.

### 4.1.2 Unequal Errors ($\Delta U \neq 0$)

We model the sensitive attribute predictor in the same way as in the previous experiment except with $\delta = c$, for a constant $c$ so that the sensitive attribute classifier still has the same total error of

---

[2]One can show that $\mathbb{E}[L(f, Y)] = \mathbb{P}(Y = 1)\mathbb{P}(Y = 0)(1 - Y_f)$ where $Y_f$ is the Youden's Index of $f$.

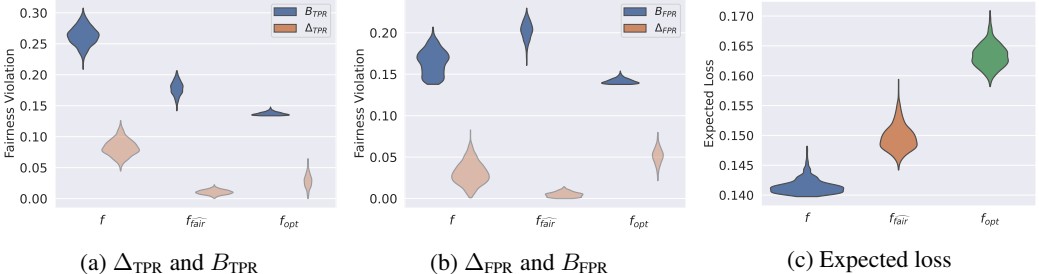

|  |  |  |
|:---:|:---:|:---:|
| (a) $\Delta_{\text{TPR}}$ and $B_{\text{TPR}}$ | (b) $\Delta_{\text{FPR}}$ and $B_{\text{FPR}}$ | (c) Expected loss |

Figure 2: Synthetic data ($\Delta U \neq 0$): true fairness violations, worst-case fairness violations, and expected loss for $f$, $f_{\widehat{\text{fair}}}$ and $f_{\text{opt}}$

$U \approx 0.04$ but distributed unevenly so that $\Delta U = -0.04$ with $U_0 = 0$ and $U_1 \approx 0.04$. As in the previous experiment we generate classifiers $f$, and perform the same correction algorithms to yield $f_{\widehat{\text{fair}}}$ and $f_{\text{opt}}$ and present the same metrics as before.

The results in Fig. 2 depict the worst-case fairness violations and expected loss for the 3 sets of different classifiers. Observe that our correction algorithm yields classifiers, $f_{\text{opt}}$, that have significantly lower $B_{\text{TPR}}$ and $B_{\text{FPR}}$. Furthermore, observe that the $f_{\widehat{\text{fair}}}$ that results from performing the naïve fairness correction algorithm in fact have higher $B_{\text{FPR}}$ the original classifier $f$! Even though the total error $U$ has remained the same, its imbalance showcases the optimality of our correction method. Lastly, observe that there is a trade-off, albeit slight, in performing our correction algorithm. The expected loss for $f_{\text{opt}}$ is higher than that of $f_{\widehat{\text{fair}}}$ and $f$.

## 4.2 Real World Data

We will now showcase our results on various real world datasets and prediction tasks. We provide a brief description of each task below and further experimental details are included in Appendix B.

**FIFA 2020 (Awasthi et al. [5])**: The task is to learn a classifier $f$, using FIFA 2020 player data [29], that determines if a soccer players wage is above ($Y = 1$) or below ($Y = 0$) the median wage based on the player's age and their overall attribute. The sensitive attribute $A$ is player nationality and the player's name is used to learn the sensitive attribute predictor $h$. We consider two scenarios, when $A \in \{\text{English}, \text{Argentine}\}$ (English = 0) and $A \in \{\text{French}, \text{Spanish}\}$ (French = 0).

**ACSPublicCoverage (Ding et al. [15])**: The task is to learn a classifier $f$, using the 2018 state census data, that determines if a low-income individual, not eligible for Medicare, has coverage from public health insurance ($Y = 1$) or does not ($Y = 0$). The sensitive attribute $A$ is sex (Female = 0) and with a separate dataset (containing the same features used to learn $f$ (disregarding sex)), we learn the sensitive attribute predictor $h$. We work with the 2018 California census data.

**CheXpert (Irvin et al. [24])**: CheXpert is a large public dataset for chest radiograph interpretation, consisting of 224,316 chest radio graphs of 65,240 patients, with labeled annotations for 14 observations (positive, negative, or unlabeled) including cardiomegaly, atelectasis, consolidation, and several others. The task is to learn a classifier $f$ to determine if an X-ray contains an annotation for *any* abnormal condition ($Y = 1$) or does not ($Y = 0$). The sensitive attribute $A$ is sex (Female = 0) and with a separate set of X-rays (different from those used to learn $f$) we learn the sensitive attribute predictor $h$.

### 4.2.1 Verification of Assumption 1

In Appendix C we provide Table 1, which demonstrates that Assumption 1 holds for the settings described above. The *Actual Value* column of Table 1 lists the rates $\widehat{\alpha}_{i,j}$ and the left and right columns list $\frac{U_i}{\widehat{r}_{i,j}}$ and $1 - \frac{U_i}{\widehat{r}_{i,j}}$ respectively. From Table 1, it is clear that the $\widehat{\alpha}_{i,j}$ lie in between $\frac{U_i}{\widehat{r}_{i,j}}$ and $1 - \frac{U_i}{\widehat{r}_{i,j}}$ as required by Assumption 1. Notice, we only list the estimated group TPRs, $\widehat{\alpha}_{i,1}$ for the ACSPublicCoverage 2018 California dataset. This is because Assumption 1, for $j = 0$, does *not* hold for this state: even though $\hat{A}$ is a very accurate predictor for $A$, the label classifier $f$ has very low estimated group FPRs, $\widehat{\alpha}_{i,0}$. As a result, our theoretical results for $\Delta_{\text{FPR}}$ cannot be used.

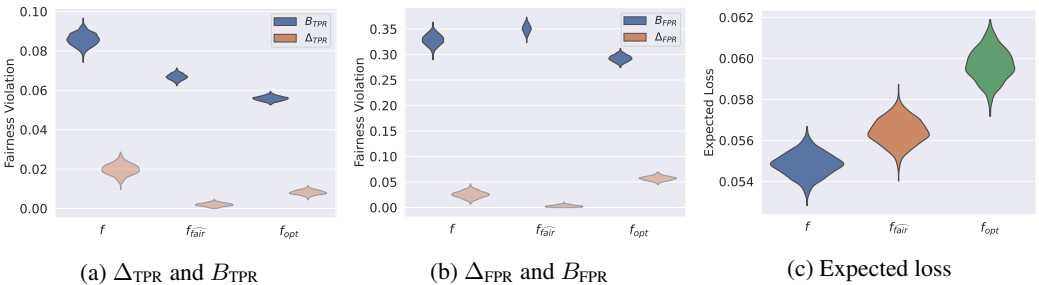

| (a) $\Delta_{\text{TPR}}$ and $B_{\text{TPR}}$ | (b) $\Delta_{\text{FPR}}$ and $B_{\text{FPR}}$ | (c) Expected loss |

Figure 3: CheXpert data: true fairness violations, worst-case fairness violations, and expected loss for $f$, $f_{\widehat{\text{fair}}}$ and $f_{\text{opt}}$

#### 4.2.2 Results on CheXpert

The task is to measure and correct for any fairness violations that $f$ may exhibit towards the sex attribute assuming we do *not* have the true sex attribute and instead have a sensitive attribute predictor, $h$. On a test dataset, we generate our predictions $\hat{Y} = f$ and $\hat{A} = h$ to yield a dataset over $(\hat{A}, Y, \hat{Y})$, for which the sex predictor, $h$, achieves an error of $U = 0.023$ with $U_0 \approx 0.008$ and $U_1 \approx 0.015$. We utilize the bootstrap method to generate 1000 samples of this dataset and for each sample, perform the same correction algorithms as before to yield $f_{\widehat{\text{fair}}}$ and $f_{\text{opt}}$ and calculate the same metrics as done in the previous experiments.

The results in Fig. 3 show that our proposed correction method performs the best in reducing $B_{\text{TPR}}$ and $B_{\text{FPR}}$. Even though $U$ is very small, since $U_1 \approx 2U_0$, simply correcting for fairness with respect to $\hat{A}$ is suboptimal in reducing the worst-case fairness violations. In particular, the results in Fig. 3 a are noteworthy as they depict how our proposed correction method and bounds allow the user to *certify* that the obtained classifier has a fairness violation in TPRs of no more than 0.06 without having access to the true sensitive attributes. Moreover, the improvement is significant, since before the correction one had $|\Delta_{\text{TPR}}| \lesssim 0.10$. In a high-stakes decision setting, such as this one where the model $f$ could be used to aid in diagnosis, this knowledge could be vital. Naturally, the expected loss is highest for $f_{\text{opt}}$ but that the increase is minimal. We make no claim as to whether this (small) increase in loss is reasonable for this particular problem setting, and the precise trade-offs must be defined in the context of a broader discussion involving policy makers, domain experts and other stakeholders.

## 5 Limitations and Broader Impacts

While our results are novel and informative, they come with limitations. First, our results are limited to EOD (and its relaxations) as definitions of fairness. Fairness is highly context-specific and in many scenarios one may be interested in utilizing other definitions of fairness. One can easily extend our results to other associative definitions of fairness, such as demographic parity, predictive parity, and others. However, extending our results to counter-factual notions of fairness [28, 11, 32] is non trivial and matter of future work. We recommend thoroughly assessing the problem and context in question prior to selecting a definition. It is crucial to ensure that the rationale behind choosing a definition is based on reasoning from both philosophical and political theory, as each definition implicitly make a distinct set of moral assumptions. For example, with EOD, we implicitly assert that all individuals with the same true label have the same effort-based utility [22]. More generally, other statistical definitions of fairness such as demographic parity and equality of accuracy can be thought of as special instances of Rawlsian equality of opportunity and predictive parity, the other hand, can be thought of as an instance of egalitarian equality of opportunity [22, 30, 3]. We refer the reader to [22] to understand the relationship between definitions of fairness in machine learning and models of Equality of opportunity (EOP) – an extensively studied ideal of fairness in political philosophy.

A second limitation of our results is Assumption 1. This assumption is relatively mild, as it is met for accurate proxy sensitive attributes (as illustrated in the chest X-rays study). Yet, we conjecture that one can do away with this assumption and consider less accurate proxy sensitive attributes with the caveat that the worst case fairness violations will no longer be linear in the TPRs and FPRs. Thus, the

characterization of the classifiers with minimal worst-case bounds would be more involved and the method to minimize these violations will likely be more difficult. Furthermore, while we proposed a simple post-processing correction method, it would be of interest to understand how one could train a classifier – from scratch – to have minimal violations. Lastly, in our setting we assume the sensitive attribute predictor and label classifier are trained on marginal distributions from the same joint distribution. As a next step, it would be important to understand how these results extend to settings where these marginal distributions come from (slightly) different joint distributions. All of this constitutes matter of future work.

Finally, we would like to remark the positive and potentially negative societal impacts of this work. Our contribution is focused on a solution to a technical problem – estimating and correcting for fairness violations when the sensitive attribute and responses are not jointly observed. However, we understand that fairness is a complex and multifaceted issue that extends beyond technical solutions and, more importantly, that there can be disconnect between algorithmic fairness and fairness in a broader socio-technical context. Nonetheless, we believe that technical contributions such as ours can contribute to the fair deployment of machine learning tools. In regards to the technical contribution itself, our results rely on predicting missing sensitive attributes. While such a strategy could be seen as controversial – e.g. because it could involve potential negative consequences such as the perpetuation of discrimination or violation of privacy – this is necessary to build classifiers with minimal worst-case fairness violations in a demographically scarce regime. On the one hand, not allowing for such predictions could be seen as one form of "fairness through unawareness", which has been proven to be an incorrect and misleading strategy in fairness [21, 10]. Moreover, our post-processing algorithm, similar to that of Hardt et al. [21], admits implementations in a differentially private manner as well, since it only requires aggregate information about the data. As a result, our method, which uses an analogous formulation with different constraints, can also be carried out in a manner that preserves privacy. Lastly, note that if one does not follow our approach of correcting for the worst-case fairness by predicting the sensitive attributes, other models trained on this data can inadvertently learn this sensitive attribute indirectly and base decisions of it with negative and potentially grave consequences. Our methodology prevents this from happening by appropriately correcting models to have minimal worst-case fairness violations.

## 6  Conclusion

In this paper we address the problem of estimating and controlling potential EOD violations towards an unobserved sensitive attribute by means of predicted proxies. We have shown that under mild assumptions (easily satisfied in practice, as demonstrated) the worst-case fairness violations, $B_{\text{TPR}}$ and $B_{\text{FPR}}$, have simple closed form solutions that are linear in the estimated group conditional probabilities $\widehat{\alpha}_{i,j}$. Furthermore, we give an exact characterization of the properties that a classifier must satisfy so that $B_{\text{TPR}}$ and $B_{\text{FPR}}$ are indeed minimal. Our results demonstrate that, even when the proxy sensitive attributes are highly accurate, simply correcting for fairness with respect to these proxy attributes might be *suboptimal* in regards to minimizing the worst-case fairness violations. To this end, we present a simple post-processing method that can correct a pre-trained classifier $f$ to yield an optimally corrected classifier, $\bar{f}$, i.e. one with with minimal worst-case fairness violations.

Our experiments on both synthetic and real data illustrate our theoretical findings. We show how, even if the proxy sensitive attributes are highly accurate, the smallest imbalance in $U_0$ and $U_1$ renders the naïve correction for fairness with respect to the proxy attributes suboptimal. More importantly, our experiments highlight our method's ability to effectively control the worst-case fairness violation of a classifier with minimal decrease in the classifier's overall predictive power. On a final observation on our empirical results, the reader might be tempted to believe that the classifier $f_{\widehat{\text{fair}}}$ (referring to, e.g., Fig. 3) is better because it provides a lower "true" fairness than that of $f_{\text{opt}}$. Unfortunately, these true fairness violations are not identifiable in practice, and all one can compute are the provided upper bounds, which $f_{\text{opt}}$ minimizes. In conclusion, our contribution aims to provide better and more rigorous control over potential negative societal impacts that arise from unfair machine learning algorithms in settings of unobserved data.

## Acknowledgments and Disclosure of Funding

This work was supported in part by NIH award R01CA287422.

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

## A  Proofs

### A.1  Explanation of Assumption 1

Assumption 1: For $i, j \in \{0, 1\}$, the classifiers $\widehat{Y} = f(X)$ and $\widehat{A} = h(X)$ satisfy

$$\frac{U_i}{\hat{r}_{i,j}} \leq \widehat{\alpha}_{i,j} \leq 1 - \frac{U_i}{\hat{r}_{i,j}}.$$

We now expand on the implications of this assumptions. Recall that $U_i = \mathbb{P}(\widehat{A} = i, A \neq i)$, $\hat{r}_{i,j} = \mathbb{P}(\widehat{A} = i, Y = j)$, and $\widehat{\alpha}_{i,j} = \mathbb{P}(\widehat{Y} = 1 \mid \widehat{A} = i, Y = j)$. Thus Assumption 1 states,

$$\mathbb{P}(\widehat{A} = i, A \neq i) \leq \mathbb{P}(\widehat{Y} = 1, \widehat{A} = i, Y = j) \leq \mathbb{P}(\widehat{A} = i, Y = j) - \mathbb{P}(\widehat{A} = i, A \neq i). \tag{10}$$

Immediately, it is clear that Eq. (10) implies

$$\mathbb{P}(\widehat{A} = i, A \neq i) \leq \mathbb{P}(\widehat{A} = i, Y = j) - \mathbb{P}(\widehat{A} = i, A \neq i) \tag{11}$$

$$\implies \mathbb{P}(\widehat{A} = i, A \neq i) \leq \frac{1}{2}\mathbb{P}(\widehat{A} = i, Y = j) \tag{12}$$

Now, the left inequality of Eq. (10) states

$$\mathbb{P}(\widehat{A} = i, A \neq i) \leq \mathbb{P}(\widehat{Y} = 1, \widehat{A} = i, Y = j) \tag{13}$$

and the right inequality of Eq. (10) states

$$\mathbb{P}(\widehat{Y} = 1, \widehat{A} = i, Y = j) \leq \mathbb{P}(\widehat{A} = i, Y = j) - \mathbb{P}(\widehat{A} = i, A \neq i)$$

which implies

$$\mathbb{P}(\widehat{A} = i, A \neq i) \leq \mathbb{P}(\widehat{A} = i, Y = j) - \mathbb{P}(\widehat{Y} = 1, \widehat{A} = i, Y = j) \tag{14}$$

$$= \mathbb{P}(\widehat{Y} = 0, \widehat{A} = i, Y = j) \tag{15}$$

If $j = 1$, then this implies

$$\mathbb{P}(\widehat{A} = i, A \neq i) \leq \mathbb{P}(\widehat{Y} = 1, \widehat{A} = i, Y = 1) \tag{16}$$

$$\mathbb{P}(\widehat{A} = i, A \neq i) \leq \mathbb{P}(\widehat{Y} = 0, \widehat{A} = i, Y = 1) \tag{17}$$

and if $j = 0$,

$$\mathbb{P}(\widehat{A} = i, A \neq i) \leq \mathbb{P}(\widehat{Y} = 1, \widehat{A} = i, Y = 0) \tag{18}$$

$$\mathbb{P}(\widehat{A} = i, A \neq i) \leq \mathbb{P}(\widehat{Y} = 0, \widehat{A} = i, Y = 0) \tag{19}$$

Any reasonable classifier $\widehat{Y}$ would have the properties

$$\mathbb{P}(\widehat{Y} = 0, \widehat{A} = i, Y = 1) \leq \mathbb{P}(\widehat{Y} = 1, \widehat{A} = i, Y = 1) \tag{20}$$

$$\mathbb{P}(\widehat{Y} = 1, \widehat{A} = i, Y = 0) \leq \mathbb{P}(\widehat{Y} = 0, \widehat{A} = i, Y = 0) \tag{21}$$

Thus, Assumption 1 is met when

$$\mathbb{P}(\widehat{A} = i, A \neq i) \leq \mathbb{P}(\widehat{Y} = j, \widehat{A} = i, Y \neq j) \tag{22}$$

## A.2 Proof of Theorem 1

We only prove the result for $|\Delta_{\text{TPR}}(f)|$ as the proof for $|\Delta_{\text{FPR}}(f)|$ is completely analogous.

*Proof.* The rules of conditional probability and the law of total probability allow us to decompose $\alpha_{1,1}$ and $\alpha_{0,1}$ in the following manner,

$$\alpha_{1,1} = \mathbb{P}(\hat{Y} = 1 \mid A = 1, Y = 1) \tag{23}$$

$$= \frac{\mathbb{P}(\hat{Y} = 1, A = 1, Y = 1)}{\mathbb{P}(A = 1, Y = 1)} \tag{24}$$

$$= \frac{\sum_{i \in \{0,1\}} \mathbb{P}(\hat{Y} = 1, A = 1, Y = 1, \hat{A} = i)}{\sum_{j \in \{0,1\}} \sum_{i \in \{0,1\}} \mathbb{P}(\hat{Y} = j, A = 1, Y = 1, \hat{A} = i)} \tag{25}$$

$$= \frac{\sum_{i \in \{0,1\}} \mathbb{P}(\hat{Y} = 1, A = 1, Y = 1 \mid \hat{A} = i) \cdot \mathbb{P}(\hat{A} = i)}{\sum_{j \in \{0,1\}} \sum_{i \in \{0,1\}} \mathbb{P}(\hat{Y} = j, A = 1, Y = 1 \mid \hat{A} = i) \cdot \mathbb{P}(\hat{A} = i)} \tag{26}$$

and

$$\alpha_{0,1} = \mathbb{P}(\hat{Y} = 1 \mid A = 0, Y = 1) \tag{27}$$

$$= \frac{\mathbb{P}(\hat{Y} = 1, A = 0, Y = 1)}{\mathbb{P}(A = 0, Y = 1)} \tag{28}$$

$$= \frac{\mathbb{P}(\hat{Y} = 1, Y = 1) - \mathbb{P}(\hat{Y} = 1, A = 1, Y = 1)}{\mathbb{P}(Y = 1) - \mathbb{P}(A = 1, Y = 1)} \tag{29}$$

$$= \frac{\mathbb{P}(\hat{Y} = 1, Y = 1) - \left[\sum_{i \in \{0,1\}} \mathbb{P}(\hat{Y} = 1, A = 1, Y = 1 \mid \hat{A} = i) \cdot \mathbb{P}(\hat{A} = i)\right]}{\mathbb{P}(Y = 1) - \left[\sum_{j \in \{0,1\}} \sum_{i \in \{0,1\}} \mathbb{P}(\hat{Y} = j, A = 1, Y = 1 \mid \hat{A} = i) \cdot \mathbb{P}(\hat{A} = i)\right]} \tag{30}$$

Therefore, $\Delta_{\text{TPR}}(f) = \alpha_{1,1} - \alpha_{0,1}$ is a function of the four probabilities given by

$$\mathbb{P}(\hat{Y} = j, A = 1, Y = 1 \mid \hat{A} = i) \tag{31}$$

which are unidentifiable in a demographically scarce regime and therefore not computable.

The Fréchet inequalities tell us that for $i, j \in \{0, 1\}$

$$\mathbb{P}(\hat{Y} = j, A = 1, Y = 1 \mid \hat{A} = i) \geq \max\{\mathbb{P}(\hat{Y} = j, Y = 1 \mid \hat{A} = i) - \mathbb{P}(A = 0 \mid \hat{A} = i), 0\} \tag{32}$$

$$\mathbb{P}(\hat{Y} = j, A = 1, Y = 1 \mid \hat{A} = i) \leq \min\{\mathbb{P}(\hat{Y} = j, Y = 1 \mid \hat{A} = i), \mathbb{P}(A = 1 \mid \hat{A} = i)\}. \tag{33}$$

Observe that, $\Delta_{\text{TPR}}(f)$ is an increasing function with respect to the two probabilities

$$\mathbb{P}(\hat{Y} = 1, A = 1, Y = 1 \mid \hat{A} = i) \tag{34}$$

and a decreasing one with respect to the two probabilities,

$$\mathbb{P}(\hat{Y} = 0, A = 1, Y = 1 \mid \hat{A} = i). \tag{35}$$

As a result, $\Delta_{\text{TPR}}(f)$ is maximal when $\mathbb{P}(\hat{Y} = 1, A = 1, Y = 1 \mid \hat{A} = i)$ achieve their maximum values and $\mathbb{P}(\hat{Y} = 0, A = 1, Y = 1 \mid \hat{A} = i)$ achieve their minimum values. On the other hand, $\Delta_{\text{TPR}}(f)$ is minimal when $\mathbb{P}(\hat{Y} = 1, A = 1, Y = 1 \mid \hat{A} = i)$ achieve their minimum values and $\mathbb{P}(\hat{Y} = 0, A = 1, Y = 1, \mid \hat{A} = i)$ achieve their maximum values. With these facts, we now provide the upper bound. Recall from Appendix A.1 that Assumption 1 implies

$$\mathbb{P}(\hat{A} = i, A \neq i) \leq \frac{1}{2}\mathbb{P}(\hat{A} = i, Y = j) \tag{36}$$

$$\mathbb{P}(\hat{A} = i, A \neq i) \leq \mathbb{P}(\hat{Y} = j, \hat{A} = i, Y \neq j) \leq \mathbb{P}(\hat{Y} = j, \hat{A} = i, Y = j) \tag{37}$$

$$\tag{38}$$

With Assumption 1 we first provide the values of $\min \left[\mathbb{P}(\hat{Y} = 0, A = 1, Y = 1, \mid \hat{A} = i)\right]$. First,

$$\mathbb{P}(\hat{Y} = 0, Y = 1, \mid \hat{A} = 1) - \mathbb{P}(A = 0 \mid \hat{A} = 1) = \frac{\mathbb{P}(\hat{Y} = 0, \hat{A} = 1, Y = 1)}{P(\hat{A} = 1)} - \frac{U_1}{\mathbb{P}(\hat{A} = 1)} \quad (39)$$

$$\geq 0 \quad (40)$$

because $\mathbb{P}(\hat{Y} = 0, \hat{A} = 1, Y = 1) - U_1 \geq 0$. Second,

$$\mathbb{P}(\hat{Y} = 0, Y = 1, \mid \hat{A} = 0) - \mathbb{P}(A = 0 \mid \hat{A} = 0) = \frac{\mathbb{P}(\hat{Y} = 0, \hat{A} = 0, Y = 1)}{P(\hat{A} = 0)} - \frac{\mathbb{P}(A = 0, \hat{A} = 0)}{\mathbb{P}(\hat{A} = 0)} \quad (41)$$

$$= \frac{\mathbb{P}(\hat{Y} = 0, \hat{A} = 0, Y = 1)}{P(\hat{A} = 0)} - \frac{\mathbb{P}(\hat{A} = 0) - U_0}{\mathbb{P}(\hat{A} = 0)} \quad (42)$$

$$= \frac{\mathbb{P}(\hat{Y} = 0, \hat{A} = 0, Y = 1)}{P(\hat{A} = 0)} - \frac{\mathbb{P}(\hat{A} = 0) - U_0}{\mathbb{P}(\hat{A} = 0)} \quad (43)$$

$$= \frac{\mathbb{P}(\hat{Y} = 0, \hat{A} = 0, Y = 1)}{P(\hat{A} = 0)} + \frac{U_0}{\mathbb{P}(\hat{A} = 0)} - 1 \quad (44)$$

Now note that,

$$\mathbb{P}(\hat{Y} = 0, \hat{A} = 0, Y = 1) = \mathbb{P}(\hat{A} = 0, Y = 1) - \mathbb{P}(\hat{Y} = 1, \hat{A} = 0, Y = 1) \quad (45)$$

$$\leq \mathbb{P}(\hat{A} = 0, Y = 1) - U_0 \quad (46)$$

where the second equality is due to Assumption 1. As a result

$$\frac{\mathbb{P}(\hat{Y} = 0, \hat{A} = 0, Y = 1)}{P(\hat{A} = 0)} + \frac{U_0}{\mathbb{P}(\hat{A} = 0)} - 1 \leq \frac{\mathbb{P}(\hat{A} = 0, Y = 1) - U_0}{P(\hat{A} = 0)} + \frac{U_0}{\mathbb{P}(\hat{A} = 0)} - 1 \quad (47)$$

$$= \frac{\mathbb{P}(\hat{A} = 0, Y = 1)}{P(\hat{A} = 0)} - 1 \leq 0 \quad (48)$$

Therefore,

$$\min \left[\mathbb{P}(\hat{Y} = 0, A = 1, Y = 1 \mid \hat{A} = 1)\right] = \mathbb{P}(\hat{Y} = 0, Y = 1, \mid \hat{A} = 1) - \mathbb{P}(A = 0 \mid \hat{A} = 1) \quad (49)$$

$$\min \left[\mathbb{P}(\hat{Y} = 0, A = 1, Y = 1 \mid \hat{A} = 0)\right] = 0 \quad (50)$$

Now we provide the values of $\max \left[\mathbb{P}(\hat{Y} = 1, A = 1, Y = 1, \mid \hat{A} = i)\right]$. First,

$$\mathbb{P}(A = 1 \mid \hat{A} = 1) = \frac{\mathbb{P}(A = 1, \hat{A} = 1)}{\mathbb{P}(\hat{A} = 1)} \quad (51)$$

$$= \frac{\mathbb{P}(\hat{A} = 1) - U_1}{\mathbb{P}(\hat{A} = 1)} \quad (52)$$

$$\geq \frac{\mathbb{P}(\hat{A} = 1, Y = 1) - U_1}{\mathbb{P}(\hat{A} = 1)} \quad (53)$$

$$\geq \frac{\mathbb{P}(\hat{A} = 1, Y = 1) - \mathbb{P}(\hat{Y} = 0, \hat{A} = 1, Y = 1)}{\mathbb{P}(\hat{A} = 1)} \quad (54)$$

$$= \frac{\mathbb{P}(\hat{Y} = 1, \hat{A} = 1, Y = 1)}{\mathbb{P}(\hat{A} = 1)} = \mathbb{P}(\hat{Y} = 1, Y = 1 \mid \hat{A} = 1) \quad (55)$$

Second,

$$\mathbb{P}(A = 1 \mid \hat{A} = 0) = \frac{\mathbb{P}(A = 1, \hat{A} = 0)}{\mathbb{P}(\hat{A} = 0)} \quad (56)$$

$$\leq \frac{\mathbb{P}(\hat{Y} = 1, \hat{A} = 0, Y = 1)}{\mathbb{P}(\hat{A} = 0)} = \mathbb{P}(\hat{Y} = 1, Y = 1 \mid \hat{A} = 0) \quad (57)$$

Therefore,

$$\max \left[ \mathbb{P}(\hat{Y} = 1, A = 1, Y = 1 \mid \hat{A} = 1) \right] = \mathbb{P}(\hat{Y} = 1, Y = 1 \mid \hat{A} = 1) \tag{58}$$

$$\max \left[ \mathbb{P}(\hat{Y} = 1, A = 1, Y = 1 \mid \hat{A} = 0) \right] = \mathbb{P}(A = 1 \mid \hat{A} = 0) \tag{59}$$

Plugging these 4 values into $\Delta_{\text{TPR}}$ will yield the upper bound,

$$B_1 + C_{0,1} = \frac{\hat{r}_{1,1}}{\hat{r}_{1,1} + \Delta U} \widehat{\alpha}_{1,1} - \frac{\hat{r}_{0,1}}{\hat{r}_{0,1} - \Delta U} \widehat{\alpha}_{0,1} + U_0 \left( \frac{1}{\hat{r}_{1,1} + \Delta U} + \frac{1}{\hat{r}_{0,1} - \Delta U} \right) \tag{60}$$

One can similarly use the assumptions to derive the lower bound,

$$B_1 - C_{1,1} = \frac{\hat{r}_{1,1}}{\hat{r}_{1,1} + \Delta U} \widehat{\alpha}_{1,1} - \frac{\hat{r}_{0,1}}{\hat{r}_{0,1} - \Delta U} \widehat{\alpha}_{0,1} - U_1 \left( \frac{1}{\hat{r}_{1,1} + \Delta U} + \frac{1}{\hat{r}_{0,1} - \Delta U} \right) \tag{61}$$

and thus $|\Delta_{\text{TPR}}| \leq \max\{|B_1 + C_{0,1}|, |B_1 - C_{1,1}|\}$. One can use same arguments to derive the upper bound for $|\Delta_{\text{FPR}}|$. $\qquad\square$

### A.3  Proof of Theorem 2

We prove the result for $|\Delta_{\text{TPR}}(f)|$. We first start by proving the existence portion of the theorem.

Let $\hat{A} = h(X)$ be a sensitive attribute classifier with errors $U_0$ and $U_1$ that produces rates $\hat{r}_{i,j} = \mathbb{P}(\hat{A} = i, Y = j)$. Let $\mathcal{F}$ be the set of classifiers for $Y$ such that $\forall f \in \mathcal{F}$, $f$ and $h$ satisfy Assumption 1. Consider any $f \in \mathcal{F}$ with group conditional probabilities, $\widehat{\alpha}_{i,j} = \mathbb{P}(\hat{Y} = 1 \mid \hat{A} = i, Y = j)$. Since we are only proving the result for $|\Delta_{\text{TPR}}(f)|$, set $j = 1$. Consider the $xy$ plane, with the $x$-axis being $\widehat{\alpha}_{0,1}$ and the $y$-axis being $\widehat{\alpha}_{1,1}$. We know,

$$\frac{U_i}{\hat{r}_{i,1}} \leq \widehat{\alpha}_{i,1} \leq 1 - \frac{U_i}{\hat{r}_{i,1}} \quad \implies \quad \frac{U_i}{\hat{r}_{i,1}} \leq \frac{1}{2}. \tag{62}$$

The two equations above define a rectangular region in the $xy$ plane with a center $(\frac{1}{2}, \frac{1}{2})$, meaning any classifier $f \in \mathcal{F}$, has $\widehat{\alpha}_{i,j}$ that are in this region.

Now, denote $\bar{\mathcal{F}}$ to be a the set of classifiers for $Y$, with group conditional probabilities $\underline{\widehat{\alpha}}_{i,1}$, that satisfy the condition,

$$\frac{\hat{r}_{0,1}}{\hat{r}_{0,1} - \Delta U} \underline{\widehat{\alpha}}_{0,1} - \frac{\hat{r}_{1,1}}{\hat{r}_{1,1} + \Delta U} \underline{\widehat{\alpha}}_{1,1} = \frac{\Delta U}{2} \left( \frac{1}{\hat{r}_{1,1} + \Delta U} + \frac{1}{\hat{r}_{0,1} - \Delta U} \right). \tag{63}$$

This condition defines a line in the $xy$ plane meaning any classifier in $\bar{\mathcal{F}}$ has $\widehat{\alpha}_{i,1}$ that are on this line. Now observe that the classifier $\bar{f} \in \bar{\mathcal{F}}$ with $\underline{\widehat{\alpha}}_{i,1} = \frac{1}{2}$, satisfy the above condition because,

$$\frac{\hat{r}_{0,1}}{\hat{r}_{0,1} - \Delta U} \left( \frac{1}{2} \right) - \frac{\hat{r}_{1,1}}{\hat{r}_{1,1} + \Delta U} \left( \frac{1}{2} \right) = \frac{1}{2} \left( \frac{\hat{r}_{0,1}}{\hat{r}_{0,1} - \Delta U} - \frac{\hat{r}_{1,1}}{\hat{r}_{1,1} + \Delta U} \right) \tag{64}$$

$$= \frac{1}{2} \left( \frac{\hat{r}_{0,1} - \Delta U + \Delta U}{\hat{r}_{0,1} - \Delta U} - \frac{\hat{r}_{1,1} + \Delta U - \Delta U}{\hat{r}_{1,1} + \Delta U} \right) \tag{65}$$

$$= \frac{1}{2} \left( 1 + \frac{\Delta U}{\hat{r}_{0,1} - \Delta U} - 1 + \frac{\Delta U}{\hat{r}_{1,1} + \Delta U} \right) \tag{66}$$

$$= \frac{\Delta U}{2} \left( \frac{1}{\hat{r}_{1,1} + \Delta U} + \frac{1}{\hat{r}_{0,1} - \Delta U} \right) \tag{67}$$

This implies that the line defined by Eq. (63) intersects the rectangular region that Assumption 1 defines. As a result, $\mathcal{F} \cap \bar{\mathcal{F}}$ is not empty, meaning there exists a classifier $\bar{f} \in \mathcal{F}$ with group conditional probabilities $\widehat{\alpha}_{i,1}$ that also satisfies the condition,

$$\frac{\hat{r}_{0,1}}{\hat{r}_{0,1} - \Delta U} \widehat{\alpha}_{0,1} - \frac{\hat{r}_{1,1}}{\hat{r}_{1,1} + \Delta U} \widehat{\alpha}_{1,1} = \frac{\Delta U}{2} \left( \frac{1}{\hat{r}_{1,1} + \Delta U} + \frac{1}{\hat{r}_{0,1} - \Delta U} \right). \tag{68}$$

Now we prove that such a classifier has minimal bounds. Theorem 1 tells us that for $f \in \mathcal{F}$

$$|\Delta_{\text{TPR}}(f)| \leq B_{\text{TPR}}(f) \triangleq \max\{|B_1 + C_{0,1}|, |B_1 - C_{1,1}|\}$$

Note that $B_1$ is linear in $\widehat{\alpha}_{1,1}$ and $\widehat{\alpha}_{0,1}$ and that $C_{0,1}$ and $C_{1,1}$ are constants such that $B_1 + C_{0,1} \geq B_1 - C_{1,1}$ simply because $B_1 + C_{0,1}$ is the upper bound for $\Delta_{TPR}$ and $B_1 - C_{0,1}$ is the lower bound. Since these bounds are affine functions shifted by a constant, then $\min \max\{|B_1+C_{0,1}|, |B_1-C_{1,1}|\}$ necessarily occurs when

$$B_1 + C_{0,1} = -B_1 - C_{1,1} \tag{69}$$

meaning

$$2B_1 = -(C_{1,1} + C_{0,1}) \tag{70}$$

have minimal upper bounds on $|\Delta_{\text{TPR}}|$. After rearranging terms, this condition is precisely

$$\frac{\hat{r}_{0,1}}{\hat{r}_{0,1} - \Delta U}\widehat{\alpha}_{0,1} - \frac{\hat{r}_{1,1}}{\hat{r}_{1,1} + \Delta U}\widehat{\alpha}_{1,1} = \frac{\Delta U}{2}\left(\frac{1}{\hat{r}_{1,1} + \Delta U} + \frac{1}{\hat{r}_{0,1} - \Delta U}\right). \tag{71}$$

## B  Experimental Details

**FIFA 2020 (Awasthi et al. [5])**: The task is to learn a classifier $f$, using FIFA 2020 player data [29], that determines if a soccer players wage is above ($Y = 1$) or below ($Y = 0$) the median wage based on the player's age and their overall attribute. The sensitive attribute $A$ is player nationality and the player's name is used to learn the sensitive attribute predictor $h$. We consider two scenarios, when $A \in \{\texttt{English}, \texttt{Argentine}\}$ ($\texttt{English} = 0$) and $A \in \{\texttt{French}, \texttt{Spanish}\}$ ($\texttt{French} = 0$). To learn the sensitive attribute predictor $h$, we train a Bidirectional Encoder Representations from Transformers (BERT) model [26] using an Adam optimizer [27] for 5 epochs. To learn the label classifier $f$, we train a Random Forest classifier. On a test dataset, we generate our predictions $\hat{Y} = f(X)$ and $\hat{A} = h(X)$ to yield a dataset over $(\hat{A}, Y, \hat{Y})$. We utilize the bootstrap method to generate 1,000 samples of this dataset and, for each sample, perform the same correction algorithms as before to yield $f_{\widehat{\text{fair}}}$ and $f_{\text{opt}}$ and calculate the same metrics as done in the synthetic data experiments. For $A \in \{\texttt{English}, \texttt{English}\}$ the nationality predictor, $h$, achieves an error of $U = 0.033$ with $U_0 \approx 0.008$ and $U_1 \approx 0.015$. For $A \in \{\texttt{French}, \texttt{Spanish}\}$ the nationality predictor, $h$, achieves an error of $U = 0.053$ with $U_0 \approx 0.025$ and $U_1 \approx 0.028$. The results for both experiments are displayed in Fig. C.1 and Fig. C.2. In addition, we include Fig. C.1d, Fig. C.1e, Fig. C.2d, and Fig. C.2e, which display the difference between $B_{\text{TPR}}(f_{\widehat{\text{fair}}})$ and $B_{\text{TPR}}(f_{\text{opt}})$ and between $B_{\text{FPR}}(f_{\widehat{\text{fair}}})$ and $B_{\text{FPR}}(f_{\text{opt}})$. These figures depict that the bounds of $f_{\text{opt}}$ are smaller than the bounds of $f_{\widehat{\text{fair}}}$ because the histograms of $B_{\text{TPR}}(f_{\widehat{\text{fair}}}) - B_{\text{TPR}}(f_{\text{opt}})$ and $B_{\text{FPR}}(f_{\widehat{\text{fair}}}) - B_{\text{FPR}}(f_{\text{opt}})$ are strictly to the right of the dashed green vertical line at 0, indicating that both differences are greater than or equal to 0.

**ACSPublicCoverage (Ding et al. [15])**: The task is to learn a classifier $f$, using the 2018 state census data, that determines if a low-income individual, not eligible for Medicare, has coverage from public health insurance ($Y = 1$) or does not ($Y = 0$). The sensitive attribute $A$ is sex ($\texttt{Female} = 0$) and with a separate dataset (containing the same features used to learn $f$ (disregarding $\texttt{sex}$)), we learn the sensitive attribute predictor $h$. We work with the 2018 California census data. To learn the sensitive attribute predictor $h$ and label classifier $f$, we train two Random Forest classifiers. On a test dataset, we generate our predictions $\hat{Y} = f(X)$ and $\hat{A} = h(X)$ to yield a dataset over $(\hat{A}, Y, \hat{Y})$. We utilize the bootstrap method to generate 1,000 samples of this dataset and, for each sample, perform the same correction algorithms as before to yield $f_{\widehat{\text{fair}}}$ and $f_{\text{opt}}$ and calculate the same metrics as done in the previous experiments. The sex predictor, $h$, achieves an error of $U = 0.07$ with $U_0 \approx 0.05$ and $U_1 \approx 0.02$. The results for this experiment are in Fig. C.3. In addition, we include Fig. C.3b, which displays the difference between $B_{\text{TPR}}(f_{\widehat{\text{fair}}})$ and $B_{\text{TPR}}(f_{\text{opt}})$. This figures depict that the TPR bounds of $f_{\text{opt}}$ are smaller than the TPR bounds of $f_{\widehat{\text{fair}}}$ because the histogram of $B_{\text{TPR}}(f_{\widehat{\text{fair}}}) - B_{\text{TPR}}(f_{\text{opt}})$ is strictly to the right of the dashed green vertical line at 0, indicating that this difference is greater than 0.

**CheXpert (Irvin et al. [24])**: CheXpert is a large public dataset for chest radiograph interpretation, consisting of 224,316 chest radio graphs of 65,240 patients, with labeled annotations for 14 observations (positive, negative, or unlabeled) including $\texttt{cardiomegaly, atelectasis, consolidation,}$

and several others. The task is to learn a classifier $f$ to determine if an X-ray contains an annotation for *any* abnormal condition ($Y = 1$) or does not ($Y = 0$). The sensitive attribute $A$ is `sex` (`Female = 0`) and with a separate set of X-rays (different from those used to learn $f$) we learn the sensitive attribute predictor $h$. To learn both $f$ and $h$ we use a DenseNet121 convolutional neural network architecture. Images are fed into the network with size $320 \times 320$ pixels. We use the Adam optimizer [27] with default $\beta$-parameters of $\beta_1 = 0.9, \beta_2 = 0.999$ and a fixed learning rate of $1 \times 10^{-4}$. Batches are sampled using a fixed batch size of 16 images and we train for 5 epochs. On a test dataset, we generate our predictions $\hat{Y} = f$ and $\hat{A} = h$ to yield a dataset over $(\hat{A}, Y, \hat{Y})$. We utilize the bootstrap method to generate 1,000 samples of this dataset, and for each sample, perform the same correction algorithms as before to yield $f_{\widehat{\text{fair}}}$ and $f_{\text{opt}}$ and calculate the same metrics as done in the previous experiments. The sex predictor, $h$, achieves an error of $U = 0.023$ with $U_0 \approx 0.008$ and $U_1 \approx 0.015$. The results for this experiment are in Fig. 3.

# C    Additional Figures & Results

Table 1: Verification of Assumption 1

| Dataset | $\widehat{\alpha}_{i,j}$ | $\frac{U_i}{\widehat{r}_{i,j}}$ | Actual Value | $1 - \frac{U_i}{\widehat{r}_{i,j}}$ |
|---|---|---|---|---|
| CheXpert | $\widehat{\alpha}_{1,1}$ | 0.03 | 0.80 | 0.97 |
| | $\widehat{\alpha}_{0,1}$ | 0.02 | 0.78 | 0.98 |
| | $\widehat{\alpha}_{1,0}$ | 0.17 | 0.22 | 0.83 |
| | $\widehat{\alpha}_{0,0}$ | 0.11 | 0.19 | 0.89 |
| FIFA 2020 (`English & Argentina`) | $\widehat{\alpha}_{1,1}$ | 0.07 | 0.93 | 0.94 |
| | $\widehat{\alpha}_{0,1}$ | 0.05 | 0.67 | 0.95 |
| | $\widehat{\alpha}_{1,0}$ | 0.13 | 0.26 | 0.87 |
| | $\widehat{\alpha}_{0,0}$ | 0.03 | 0.05 | 0.97 |
| FIFA 2020 (`French & Spanish`) | $\widehat{\alpha}_{1,1}$ | 0.08 | 0.86 | 0.92 |
| | $\widehat{\alpha}_{0,1}$ | 0.13 | 0.83 | 0.87 |
| | $\widehat{\alpha}_{1,0}$ | 0.11 | 0.13 | 0.89 |
| | $\widehat{\alpha}_{0,0}$ | 0.10 | 0.10 | 0.95 |
| ACSPublicCoverage (California) | $\widehat{\alpha}_{1,1}$ | 0.13 | 0.39 | 0.87 |
| | $\widehat{\alpha}_{0,1}$ | 0.24 | 0.33 | 0.76 |

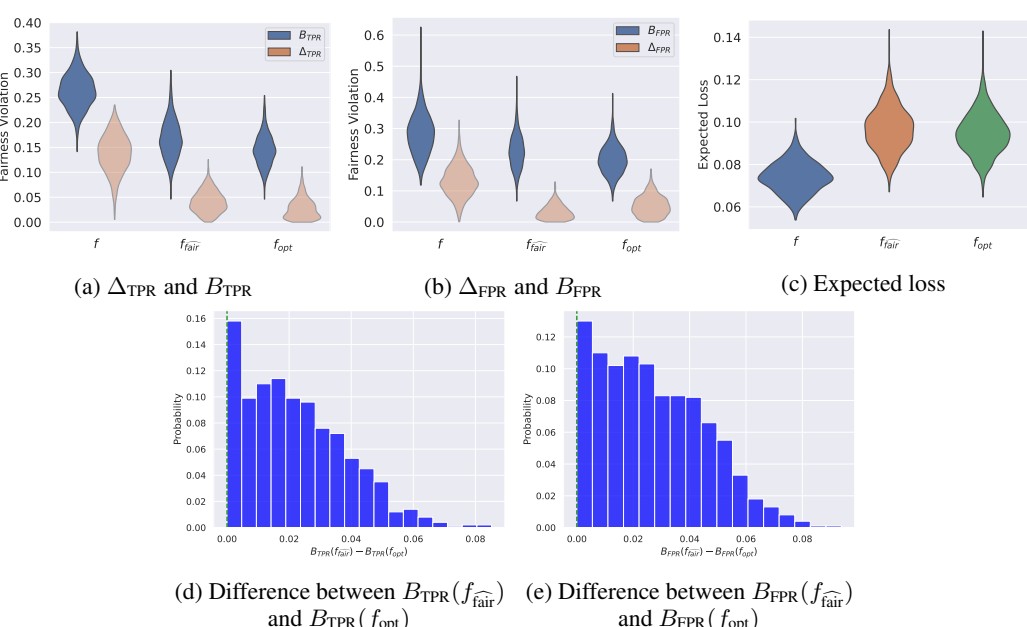

(a) $\Delta_{\text{TPR}}$ and $B_{\text{TPR}}$

(b) $\Delta_{\text{FPR}}$ and $B_{\text{FPR}}$

(c) Expected loss

(d) Difference between $B_{\text{TPR}}(f_{\widehat{\text{fair}}})$ and $B_{\text{TPR}}(f_{\text{opt}})$

(e) Difference between $B_{\text{FPR}}(f_{\widehat{\text{fair}}})$ and $B_{\text{FPR}}(f_{\text{opt}})$

Figure C.1: Results for FIFA 2020 dataset with $A \in \{$`English`, `Argentine`$\}$.

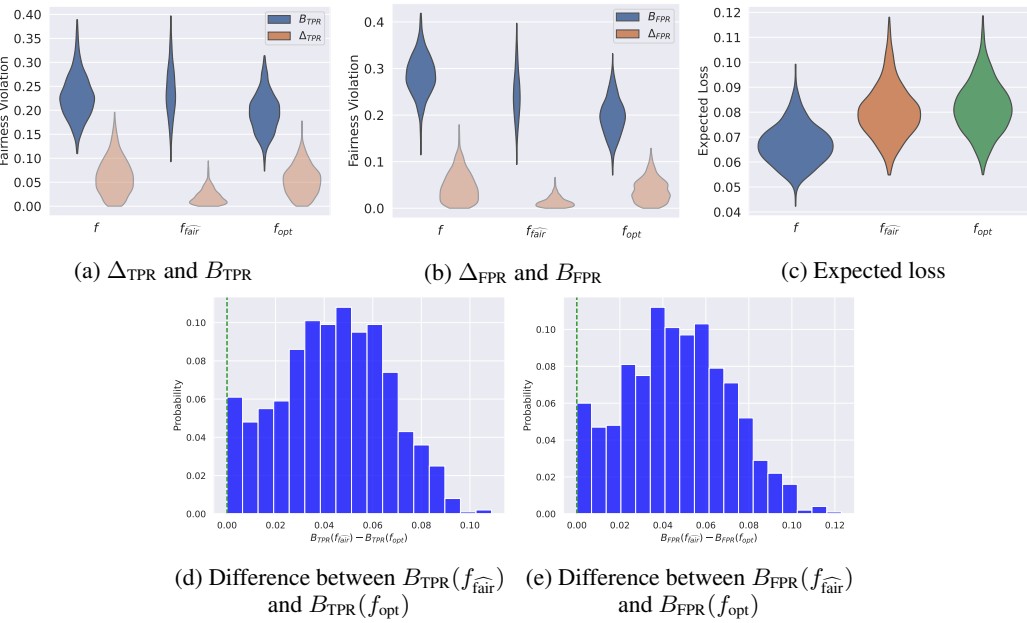

(a) $\Delta_{\text{TPR}}$ and $B_{\text{TPR}}$

(b) $\Delta_{\text{FPR}}$ and $B_{\text{FPR}}$

(c) Expected loss

(d) Difference between $B_{\text{TPR}}(f_{\widehat{\text{fair}}})$ and $B_{\text{TPR}}(f_{\text{opt}})$

(e) Difference between $B_{\text{FPR}}(f_{\widehat{\text{fair}}})$ and $B_{\text{FPR}}(f_{\text{opt}})$

Figure C.2: Results for FIFA 2020 dataset with $A \in \{\texttt{French}, \texttt{Spanish}\}$.

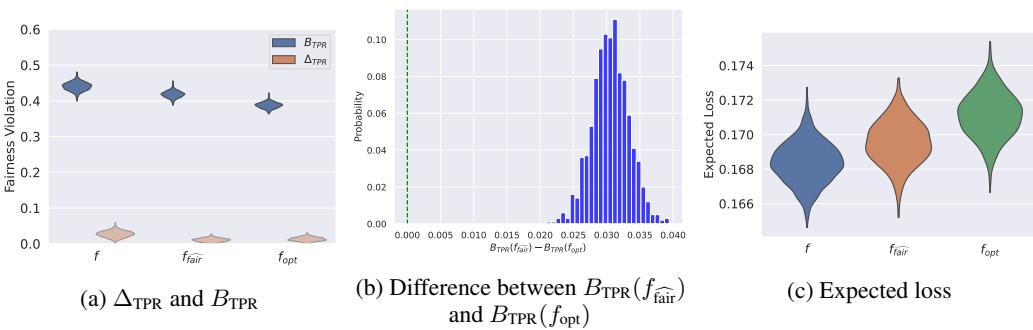

(a) $\Delta_{\text{TPR}}$ and $B_{\text{TPR}}$

(b) Difference between $B_{\text{TPR}}(f_{\widehat{\text{fair}}})$ and $B_{\text{TPR}}(f_{\text{opt}})$

(c) Expected loss

Figure C.3: Results for ACSPublicCoverage 2018 California census data

