# OpenReview forum: "Estimating and Controlling for Equalized Odds via Sensitive Attribute Predictors"
_NeurIPS.cc/2023/Conference — NeurIPS 2023 poster_

### Official Review · Reviewer_ntnH · 2023-06-09

**Soundness:** 4 excellent
**Presentation:** 4 excellent
**Contribution:** 3 good
**Rating:** 7
**Confidence:** 4

**Summary:**

This work provides tight and computable bounds on the EOD violation of a classifier in a setting where the sensitive attributes are unknown. In addition, a post-processing technique is proposed to provably yield classifiers that maximize prediction power while achieving minimal worst-case EOD violations with respect to unobserved sensitive attributes. Experiments are done on both synthetic and real data.

**Strengths:**

Significance: The problem under study is interesting and important. It would be useful to be able reduce the upper bound of certain fairness violation in models when sensitive attributes are unknown.

Originality and quality: the proposed solution is novel and technically sound.

Clarity: the presentation is clear with limitations analyzed in details.

**Weaknesses:**

1. One assumption of the proposed approach is that there exists a dataset of (X,A) to learn $\hat{A} = h(X)$ from. This could limit the use of the approach and should be discussed.

2. It would be beneficial to demonstrate the verification of Assumption 1 on the two datasets.


**Questions:**

Q1: can you elaborate on the potential bad outcome of applying $f_{opt}$ yields a less accurate and more biased (in terms of $\Delta_{FPR}$) model as in Figure 3?

Q2: Is that possible to use a different dataset of similar features (but not the same features as in the experiments) to learn $\hat{A} = h(X)$ (in some form of transfer learning)?

**Limitations:**

Results on the CheXpert data shows that, although the proposed method successfully reduces the worst case scenario TPR and FPR, it may actually increase the true TPR or FPR. Figure 3b shows an increase of $\Delta_{FPR}$ of $f_{opt}$ over $f$. This should be discussed as the potential outcome of applying $f_{opt}$ can be a less accurate and more biased model.

---

> ### Author Rebuttal · Authors · 2023-08-09
>
> **Comment/Concern 1: Assumption of the proposed approach is that there exists a dataset of $(X,A)$ to learn $\widehat{A} = h(X)$. This could limit the use of the approach and should be discussed**
>
> Answer: We completely agree. All the contributions of our work rely on there being a dataset of $(X,A)$ to learn $\widehat{A} = h(X)$. As we explain in the Introduction, this is a milder assumption (or often alternative assumption) to having a dataset from the entire joint distribution over $(X,Y,A)$. Our results are limited to such cases, however, so they do not apply when a predictor $\widehat{A}$ for $A$ cannot be constructed.
>
> **Comment/Concern 2: Verification of Assumption 1 on the two datasets**
>
> Answer: We thank the reviewer for mentioning that we do not formally demonstrate that Assumption 1 holds on the datasets used in our experiments.  We refer the reviewer to the global author rebuttal where there is **pdf** attached. In this pdf we provide Table 1 that demonstrates that Assumption 1 (or its relaxations) holds for the datasets in the paper along with new data sets (descriptions of the new datasets/experiments are detailed in the global author rebuttal).
>
> **Comment/Concern 3: Can you elaborate on the potential bad outcome of applying $f_{opt}$ yields a less accurate and more biased**
>
> Answer: Certainly. The reviewer is correct: it it clear from Figure 3 that applying $f_{opt}$ yields a less accurate and more biased (in terms of $\Delta_{FPR}$) model. Using such a model could thus be seen as controversial because it might perpetuating unfairness. However, in some settings, one may have the goal of constructing a classifier, $f$, that has a fairness violation that does not exceed a budget $\epsilon$ (for example, as required by regulatory agencies). In these cases, our method allows one to potentially achieve this goal by reducing the worst-case violation so that it is less than $\epsilon$.  Nonetheless, we recommend thoroughly assessing the problem at hand with domain experts to assess if the potential (small) bias introduced by our method constitutes a manageable risk. This is because in a demographically scarce setting one cannot calculate the true fairness violation, and thus, one cannot have an idea of how the true violation ultimately changes as $f$ is modified.
>
> **Comment/Concern 4: Is it possible to use a different dataset of similar features (but not the same features as in the experiments) to learn $\widehat{A} = h(X)$ (in some form of transfer learning)**
>
> Answer: In some restricted sense, yes: one could learn a $\widehat{A} = h(X)$ from $\mathcal{D}_1$ using whatever features that are available to the user. However, these features must be present in the dataset $\mathcal{D}_2$ over $(X,Y)$ that is used to learn $\widehat{Y} = f(X)$. If this is not the case, then the user with access to $\mathcal{D}_2$, would not be able to impute proxy sensitive attributes $\widehat{A}$ in lieu of $A$ and compute the bounds on the true fairness violations. However, the features in $\mathcal{D}_1$ might include more features than those present in $\mathcal{D}_2$

---

> > ### Comment · Reviewer_ntnH · 2023-08-16
> >
> > Thanks for the authors' revision. I would recommend accepting this paper.

---

### Official Review · Reviewer_RpWQ · 2023-07-01

**Soundness:** 3 good
**Presentation:** 4 excellent
**Contribution:** 3 good
**Rating:** 7
**Confidence:** 4

**Summary:**

The paper explores the fairness problem in machine learning when sensitive attributes (e.g., demographics and gender) are unavailable for practitioners. The authors focus on the Equalized-Odds (EOD), a well-known definition of fairness for classifiers. The authors provide bounds for EOD violation in a setting where sensitive attributes are unavailable. Additionally, they propose a post-processing method to control the worst-case EOD. Finally, the authors illustrate their results via synthetic and real-world datasets experiments.

**Strengths:**

* The paper is very well written. Particularly the related work section is very informative.

* The authors explore an interesting problem for the community and propose a different view of the problem, i.e., a worst-case optimization.

* The theoretical guarantees provide a method for practitioners to access the quality of the method (assuming a worst-case scenario).

* The proposed method is simple, intuitive, and computationally inexpensive.


**Weaknesses:**

* The evaluation in the main paper is limited. The authors only apply their method to one real-world example. I suggest adding at least two more examples and as many examples as possible in the appendix. These examples can convince us that: (i) Assumption 1 is followed in multiple scenarios and (ii) your method helps to improve fairness.

*  The fact that the Naïve approach had a better "true" EOD is interesting, and it might indicate that your method is not very efficient, i.e., using the sharp upper bound as a proxy for the true value is too pessimistic. Performing more experiments in real-world datasets can help the authors and readers understand when minimizing the upper bound is better than the naïve approach.

* It needs to be clarified how to extend the approach to a setting where the sensitive attribute is not binary. In the introduction, the authors list examples of sensitive attributes as "demographics and gender," which are not binary. A subsection is recommended to discuss how your results would change when $A$ is not binary.


**Questions:**

* Is it possible to extend this worst-case analysis to other post-processing methods? Combining it with the optimized method proposed in [1] would be very interesting.

* In Theorem 2 the authors assume that $\mathcal{F}$ is such that $ \forall f \in \mathcal{F}$ the Assumption 1 is followed. After that, they conclude that then, there exists a fair model in $\mathcal{F}$. This doesn’t seem correct. For example, take $\mathcal{F} = \{f_0\}$ where $f_0$ follows Assumption 1 but is not worst-case optimal. Then, there is no fair model in $\mathcal{F}$. Perhaps, the authors mean $\mathcal{F}$ to be the set of *all* models that follow Assumption 1. However, in this case, assuming knowledge of $\mathcal{F}$ does not sound reasonable. Could the authors clarify this point?

* Could you describe (in the main paper) when the worst-case bound is achieved?


**Limitations:**

The authors discuss their work limitations. I also suggest discussing the ethics of predicting users' sensitive attributes.

---

> ### Author Rebuttal · Authors · 2023-08-10
>
> **Comment/Concern 1: More real-world experiments to convince us that Assumption 1 is followed in many scenarios and your method helps to improve fairness.**
>
> Answer: Thank you for the suggestion. We have performed additional experiments to show that Assumption 1 holds in multiple scenarios and showcase the utility of our results. We refer the reviewer to the global author rebuttal which provides details of the experiments and an attached **pdf** which contain the results.
>
> We would also like to comment on the point the reviewer made that our method "helps to improve fairness". It actually improves the worst-case fairness violation and we make no claim that it reduces the true violation. As we explain in our paper, doing so is impossible in general: one cannot claim that performing any correction of $\hat Y$ with respect to $\hat A$ provably reduces the true violation unless strict and unverifiable assumptions are made about $\hat A$ and $\hat Y$ (See Awasthi et al. [2021] and Kallus et al. [2022] for further details)
>
> **Comment/Concern 2: Naïve approach had a better true EOD**
>
> We thank the reviewer for bringing this up. On one hand, improvements on the real fairness are not verifiable (as explained above). On the other hand, Awasthi et al. [2019] study the properties of a classifier $\tilde{Y}$ that comes from using the post-processing method by Hardt et al. [2016] with noisy sensitive attributes $\hat A$ in lieu of $A$ (i.e. the "naive" approach). They provide conditions on $\hat A$ that are necessary for $\tilde{Y}$ to have a smaller true fairness violation than the original classifier, $\hat{Y}$. One is that $\hat{Y} \perp \hat{A} \mid A,Y$, which is strict and not verifiable in demographically scarce regimes. So when this condition does not hold, no guarantees exist that show that correcting with respect to $\hat{A}$ reduces the true fairness violation. While our post-processing method also does not provide any guarantee about the true violation, it is guaranteed to provably reduce the worst-case violation.
>
> **Comment/Concern 3: Extending the result to settings with non binary sensitive attributes**
>
> We agree that cases where $A$ is not binary are interesting. Definitions of fairness can be extended to these situations. For simplicity, suppose one wants to enforce Equal Opportunity and $A \in \{1, \dots, k\}$. Then one can require $P(\widehat{Y} = 1 \mid A = i, Y = 1) = P(\widehat{Y} = 1 \mid A = j, Y = 1)$ for all $i \neq j$. Enforcing this using the method in Hardt et al. [2016] would mean including $\frac{k(k-1)}{2}$ constraints. In our setting, to reduce the worst-case violations using $\hat{A}$ we would also have to include $\frac{k(k-1)}{2}$ constraints since there are that many bounds to reduce. While this can be done, it will naturally result in the expected loss to increase since there are more constraints.
>
> **Comment/Concern 4:  Extending worst-case analysis to other post-processing methods/Combining it with the optimized method in [1].**
>
> In Theorem 2 we provide necessary conditions for $\hat Y$ to have minimal worst-case fairness violations. As long we can add these constraints to other post-processing methods, we believe we can combine our analysis with other methods. The reviewer mentions a method proposed in [1] without a citation. We will gladly answer any questions about combining our results with this method once the reviewer clarifies which work they are referencing.
>
> **Comment/Concern 5: Clarity of Theorem 2**
>
> We thank the reviewer for pointing out this inaccuracy in our original statement of Theorem 2 -- we meant that $\mathcal F$ simply parametrizes estimators for the random variable $Y$. Here is the rewritten version of the result in its correct form.
>
> Theorem 2: Let $\hat{A} = h(X)$ be a fixed sensitive attribute classifier with errors $U_0$ and $U_1$ that produces rates $\hat{r}\_{i,j} = P(\hat{A} = i, Y = j)$. Let $\mathcal{F}$ be the set of all predictors of $Y$ parametrized by the rates $\hat{r}\_{i,j}$ that, paired with $h$, satisfy Assumption 1. Then, $\exists \overline{Y} \in \mathcal{F}$ with group conditional probabilities, $\widehat{\underline{\alpha}}_{i,j} =  P(\overline{Y} = 1 \mid \hat{A} = i, Y = j)$ that satisfy the following condition,
>
> $$\frac{\hat{r}\_{0,j}}{\hat{r}\_{0,j} - \Delta U}\widehat{\underline{\alpha}}\_{0,j} - \frac{\hat{r}\_{1,j}}{\hat{r}\_{1,j} + \Delta U}\widehat{\underline{\alpha}}\_{1,j} = \frac{\Delta U}{2}\\left(\frac{1}{\hat{r}\_{1,j} + \Delta U} + \frac{1}{\hat{r}\_{0,j} - \Delta U}\\right)$$
>
> Furthermore, any such $\overline{Y}$ has minimal maximal-fairness violation:
>
> $$|\Delta\_{TPR}(\overline{Y})| \leq B\_{TPR}(\overline{Y})  \leq {B}_{TPR}(\hat{Y}) \quad \text{and} \quad |\Delta\_{FPR}(\overline{Y})| \leq B\_{FPR}(\overline{Y}) \leq {B}\_{FPR}(\hat{Y})$$
>
> This states that any predictor of $Y$ that satisfies Assumption 1 AND the condition above, has a worst case fairness violation that is less than or equal to any predictor of $Y$ that only satisfies Assumption 1. In other words, this condition is a necessary condition for the worst case fairness violation to be minimal.
>
> **Comment/Concern 5: Could you describe when the worst-case bound is achieved?**
>
> Answer: Certainly, we will add this in the revised version of the paper. Under Assumption 1, for $\Delta_{TPR}$, the bound is achieved when for $i \in \\{0,1\\}$ the unobserved quantities $P(\hat{Y} = 1, A = 1, Y = 1 \mid \hat{A} = i)$ and $P(\hat{Y} = 0, A = 1, Y = 1 \mid \hat{A} = i)$ achieve their feasible maximal and minimal values respectively. To be more precise, when:
> $$P(\hat{Y} = 1 , A = 1 , Y = 1 \mid \hat{A} = 1) = P(\hat{Y} = 1 , Y = 1 \mid \hat{A} = 1)$$
> $$P(\hat{Y} = 1 , A = 1 , Y = 1 \mid \hat{A} = 0) = P(A = 1 \mid \hat{A} = 0)$$
> $$P(\hat{Y} = 0 , A = 1 , Y = 1 \mid \hat{A} = 1) = P(\hat{Y} = 0 , Y = 1 \mid \hat{A} = 1) - P(A = 0 \mid \hat{A} = 1)$$
> $$P(\hat{Y} = 0 , A = 1 , Y = 1 \mid \hat{A} = 0) = 0$$
>
> A similar result holds for $\Delta\_{FPR}$

---

> > ### Comment · Reviewer_RpWQ · 2023-08-14
> >
> > I want to thank the authors for their careful answers. The paper substantially improved after the revision.
> >
> > I am suggesting that the papers get accepted, increasing the score to 7 and the contribution score to 3.
> >
> > Sorry for not sharing the citation. I probably had a problem with the format I copied. However, I still think it could be interesting for the authors to explore the connections between [1] and their work.
> >
> > [1] Alghamdi, W. et al. Beyond Adult and COMPAS: Fair Multi-Class Prediction via Information Projection. NeurIPS 2022.

---

### Official Review · Reviewer_XmGE · 2023-07-05

**Soundness:** 3 good
**Presentation:** 4 excellent
**Contribution:** 3 good
**Rating:** 7
**Confidence:** 3

**Summary:**

The paper proposes a tight upper bounds for the equalized odds violation of a predictor in a setting without sensitive attributes.
It also presents a post-processing correction method to control the worst-case equalized odds violation and presents results on a variety of synthetic and real datasets.


**Strengths:**

1. The paper addresses a critical issue in fair ML.

2. The tight and computable upper bounds for equalized odds (EOD) violation is a valuable contribution as it allows for a precise understanding of the worst-case EOD violation of a predictor.

3. The proposed post-processing correction method for controlling worst-case EOD is interesting.

4. The paper is backed by experiments on both synthetic and real datasets, strengthening the validity of the results.

**Weaknesses:**

1. Assumption 1 may limit its applicability in scenarios where the proxy sensitive attributes are not accurate.

2. The paper's results are limited to EOD and its relaxations as definitions of fairness. Extending the results to other notions of fairness may be non-trivial and deserves further consideration.

3. The paper does not consider how one could train a classifier from scratch to have minimal violations, which could be an important


**Questions:**

1. Could you provide more insight into the limitations of Assumption 1 and how these might be addressed?

2. Could you elaborate on how the proposed post-processing correction method could be implemented in real-world scenarios?

3. How could your results be extended to other definitions of fairness, and what challenges might be encountered in this process?



**Limitations:**

The paper relies on Assumption 1, which may limit its applicability in scenarios where the proxy sensitive attributes are not accurate.
Despite this limitation, this work is a nice contribution.

---

> ### Author Rebuttal · Authors · 2023-08-09
>
> **Comment/Concern 1: Limitations of Assumption 1 and how they might be addressed**
>
> Answer: Absolutely. For clarity, we re-write Assumption 1 below:
>
> For $i,j \in \\{0,1\\}$  the classifiers $\hat{Y} = f(X)$ and $\hat{A} = h(X)$, with errors $U_i = P(\hat A = i, A \neq i)$, rates $r_{i,j} = P(\hat A = i, Y = j)$, and conditional errors $\hat {\alpha}\_{i,j} = P(\hat Y = 1 | \hat{A}=i, Y = j )$ satisfy
>
> $$\frac{U_i}{\hat{r}\_{i,j}} \leq {\hat \alpha}\_{i,j} \leq 1 - \frac{U_i}{\hat{r}\_{i,j}}$$
>
> In short, this assumption says that the estimated-group conditional errors of $\hat A$ are not higher than (relatively) to the errors of $\hat Y$. Naturally, all works that focus on using proxy sensitive attributes to measure fairness make some assumptions, and our work is no different in this regard. However, our assumption is significantly milder than those in existing results. For example, [Awasthi et al., 2021] assume that $\hat{Y} \perp \hat{A} \mid A,Y$,  which is strict and not verifiable in demographically scarce regimes. Our assumption is directly verifiable in practice, and does not require independence -- it simply requires $\hat{A}$ to be reasonably accurate. In fact, it has been shown in many settings that accurate $\hat{A}$ can be developed [Baines and Courchane, 2014, Elliott et al., 2009, Imai and Khanna, 2016, Gichoya et al., 2022], as required by Assumption 1. Lastly, we would like to point out that if Assumption 1 holds for $i \in \\{0,1\\}$ and $j = 1$ (only), then all the results for $\Delta_{TPR}$ hold (similarly for $j = 0$, and $\Delta_{FPR}$). Thus, if one only cares about the equal opportunity definition of fairness, the assumption only needs to hold for $i \in \\{0,1\\}$ and $j = 1$ and one only needs to add this constraint to the post-processing algorithm.
>
> Limitations: When the features $X$ are not predictive of $A$, yielding a poor $\hat{A}$, Assumption 1 may not hold. In these settings, we can still derive a tight upper bounds. However, they are no longer linear in the group estimated TPRs and FPRs. Thus, the characterization of classifiers with minimal worst-case bounds is more involved and the methods to minimize these violations will likely not be as succinct and elegant as the results we presented. We will include this comment in the revised version of our manuscript.
>
> Lastly, we performed additional experiments to demonstrate Assumption 1 holds in various settings. We refer the reviewer to the global author rebuttal for the details of these experiments along with results in the attached **pdf**.
>
> **Comment/Concern 2: Post-processing method in the real world**
>
> Answer: We envision our method being used in a scenario where one does not have access to $A$ but requires $\hat{Y}$ to have an Equalized Odds (or its relaxations) violation no greater than a budget $\epsilon$ (for example, as determined by regulatory agencies). In this scenario, with the use of $\hat{A}$ and our Theorem 1, one can still compute the worst case fairness violation of $\hat{Y}$. Furthermore, by means of Theorem 2 and our post-processing algorithm, one can reduce this quantity. Note, this provides a provable certificate, since if the worst case fairness violation is less than epsilon, so is the true fairness violation.
>
> **Comment/Concern 3: Extension to other definitions of fairness**
>
> Answer: Our results apply to equalized odds, equal opportunity, predictive equality, and can be trivially extended to demographic parity. While this is a restriction, note that this is a main set of fairness definitions considered in many previous works. Extending our results to other definitions, such as causal or individual fairness, is challenging for a variety of reasons.
>
> For example, in causal fairness, a key quantity of interest is the average treatment effect (ACE), $E[Y(A=1)] - E[Y(A=0)]$, where $Y(A)$ is a counterfactual quantity. Under certain assumptions on the data generating process, techniques from causal inference can be used to identify the ACE [Pearl, 2009]. However, in all such scenarios, the treatment $A$ (i.e. the sensitive attribute) is **observed**. In our setting, $A$ is not observed, and so the way to apply techniques from causal inference remains an unclear but interesting research direction.
>
> As another example, considering individual fairness requires that "similar" individuals are classified "similarly" by $\hat{Y}$. Similarity is measured with respect to a fair metric, $d_x$. If $d_x$ does not involve $A$, then one can easily check if individual fairness holds. If $d_x$ does involve $A$, then evaluating individual fairness relies on how $d_x$ changes when proxies $\hat{A}$ are used in place of $A$. These are indeed interesting research avenues that will constitute future work.
>
> **Comment/Concern 4: Training a classifier from scratch to have minimal worst-case fairness violations**
>
> Answer: Empirically, one could do this by an in-processing training method where one is trying to minimize the expected loss of a classifier with the constraints we provide in Assumption 1 and Theorem 2. In doing this, in principle, one would end up with a classifier $\tilde{Y}$ with minimal worst-case fairness violations and expected loss, potentially smaller than that of classifier $\overline{Y}$ returned from our post-processing algorithm. This is because in the post-processing algorithm, the set of classifiers one is optimizing over is constrained by the initial predictor $\hat{Y}$, meaning $\overline{Y}$ cannot have smaller expected loss than $\hat{Y}$. The in-training optimization problem would have no such restriction. However, note that our post-processing algorithm is computationally inexpensive and provably optimal, whereas the in-processing counterpart could be computationally expensive and not be necessarily guaranteed (e.g. it would result in an non-convex optimization problem if the classifier is a deep neural net). We will add these discussions to the revised version of our manuscript.

---

> > ### Comment · Reviewer_XmGE · 2023-08-11
> >
> > Thanks for the detailed rebuttal. All my questions have been addressed.

---

### Author Rebuttal · Authors · 2023-08-10

We would like to thank all the reviewers for the insightful questions about our work. All the reviewers had questions about and wanted to see a verification of Assumption 1. For clarity, we re-write Assumption 1 below:

For $i,j \in \\{0,1\\}$  the classifiers $\widehat{Y} = f(X)$ and $\widehat{A} = h(X)$, with errors $U_i = P(\hat A = i, A \neq i)$, rates $r_{i,j} = P(\hat A = i, Y = j)$, and conditional errors $\hat {\alpha}\_{i,j} = P(\hat Y = 1 | \hat{A}=i, Y = j )$ satisfy

$$\frac{U_i}{\hat{r}\_{i,j}} \leq {\hat \alpha}\_{i,j} \leq 1 - \frac{U_i}{\hat{r}\_{i,j}}$$

We first would like to point out that the assumption can be relaxed, meaning, if it holds for $i \in \\{0,1\\}$ and for $j = 1$ (only), then all the results for $\Delta_{TPR}$ still hold. Similarly, if these hold for $j = 0$, the results for $\Delta_{FPR}$ hold true. As a result, if one only cares about the equal opportunity definition of fairness, one only needs a relaxation of the assumption to hold (it just needs to hold for  $i \in \\{0,1\\}$ and $j = 1$) and only this constraint needs to be added to the post-processing algorithm.

In the attached **pdf**, the results in Table 1 verifies that Assumption 1 (or its relaxations) indeed holds for various datasets. We also present Figures 1 and 2, which extend our results for these new datasets.  We will discuss the table and figures in more detail, but we first provide a brief summary of the new experiments below:

**FIFA Experiment** (Adopted from Awasthi et al. [2021]): We use the FIFA 20 player dataset and aim to predict whether a soccer player’s wage is above ($Y = 1$) or below ($Y = 0$) the median wage based on the player’s age and their overall attribute. We consider nationality as the sensitive attribute $A$ and use the player's name to predict this attribute. Assumption 1 holds for various pairs of nationalities and in Table 1 we demonstrate this for the Argentine/English and France/Spain pairs.

**ACSPubCov** Experiment (Adopted from Ding et al. [2021]): The task is to predict, using 2018 state census data, whether a low-income individual, not eligible for Medicare, has coverage from public health insurance . We consider sex to be sensitive attribute $A$. We determine that for 28 out of 50 states, Assumption 1 holds for $\Delta_{TPR}$ while the assumption for $\Delta_{FPR}$ does not hold for any. We showcase that Assumption 1 holds for the state of Illinois.

We now discuss the figures and tables.

**Table 1**

We provide Table 1, which demonstrates that Assumption 1 (or its relaxations) holds for a variety of experiments/datasets. The *Actual Value* column of Tables 1a and 1b lists the ${\hat \alpha}\_{i,j}$ and the left and right columns list $\frac{U_i}{\hat{r}\_{i,j}}$ and $1 - \frac{U_i}{\hat{r}\_{i,j}}$ respectively for all the datasets. From Table 1, it is clear that the ${\hat \alpha}\_{i,j}$ lie in between $\frac{U_i}{\hat{r}\_{i,j}}$ and $1 - \frac{U_i}{\hat{r}\_{i,j}}$ as is required by Assumption 1. Notice, we only list the estimated group TPRs, ${\hat \alpha}\_{i,1}$ for the ACSPubCov Illinois dataset. This is because, as mentioned, Assumption 1 fails to hold for $\Delta_{FPR}$ for all 50 states. The reason for this is that, while $\widehat{A}$ is indeed very accurate (e.g. $U \approx 0.08$ on the Illinois data), the label classifier $\widehat{Y}$ has very low estimated group FPRs ${\hat \alpha}\_{i,0}$.

In conclusion, these new results show that our assumption holds in several real datasets and cases, while also showcasing when (and why) it doesn't.

**Figures 1 and 2**

In Figure 1, we present the results for the FIFA 2020 player dataset restricted to English and Argentine nationalities. Similar to the experiments in the manuscript, we learn a sensitive attribute predictor $\hat{A} = h(X)$ which achieves an error of $U = 0.025$ with $U_1 \approx 0.02$ and $U_0 \approx 0.005$ and we learn a label classifier $\hat{Y} = f(X)$. On a test data set, we generate our predictions $\hat{Y} = f$ and $\hat{A} = h$ to yield a dataset over $(\hat{A}, Y, \hat{Y})$ and utilize the bootstrap method to obtain to generate 1000 samples from this dataset and for each sample, perform the correction algorithms to yield $f_{\widehat{\textrm{fair}}}$ and $f_{\textrm{opt}}$. Figure 1a and 1b display $B\_{TPR}$ and $\Delta\_{TPR}$ (not identifiable) for the various classifiers. We additionally add Figure 1d and 1e which are normalized histograms constructed from the bootstrapped samples of $B\_{TPR}(f_{\widehat{\textrm{fair}}}) - B\_{TPR}(f_{\textrm{opt}})$ and $B\_{FPR}(f_{\widehat{\textrm{fair}}}) - B\_{FPR}(f_{\textrm{opt}})$. Observe how both quantities are greater than or equal to 0, indicating that $B\_{TPR}(f_{\textrm{opt}})$ and $B\_{FPR}(f_{\textrm{opt}})$ are smaller than $B\_{TPR}(f_{\widehat{\textrm{fair}}})$ and $B\_{FPR}(f_{\widehat{\textrm{fair}}})$ respectively. This indicates that our post-processing method is better than simply correcting with respect to $\hat A$ in regards to reducing the worst-case fairness violation (in fact our method is optimal as noted in Theorem 2). Lastly, Figure 1c shows that the expected loss increases for $f_{\textrm{opt}}$ but that the increase is minimal.

We perform the same experimental process on the ACSPubCov Illinois dataset and the results for the TPR quantities are depicted in Figure 2. Note we do not show FPR related quantities because Assumption 1 fails to hold.

---

### Decision · Program_Chairs · 2023-09-21

**Decision:**

Accept (poster)

**Comment:**

After some discussion, all reviewers agreed that the paper should be accepted. The authors should incorporate their reply to the reviewers that further justified Assumption 1.